# Fast Interactive Search under a Scale-Free Comparison Oracle

**Daniyar Chumbalov**[* †1]     **Lars Klein**[*1]     **Lucas Maystre**[2]     **Matthias Grossglauser**[1]

[1]INDY Lab, EPFL, Lausanne, Switzerland
[2]Spotify, London, UK

## Abstract

A comparison-based search algorithm lets a user find a target item $t$ in a database by answering queries of the form, "Which of items $i$ and $j$ is closer to $t$?" Instead of formulating an explicit query (such as one or several keywords), the user navigates towards the target via a sequence of such (typically noisy) queries. We propose a scale-free probabilistic oracle model called $\gamma$-CKL for such similarity triplets $(i, j; t)$, which generalizes the CKL triplet model proposed in the literature. The generalization affords independent control over the discriminating power of the oracle and the dimension of the feature space containing the items. We develop a search algorithm with provably exponential rate of convergence under the $\gamma$-CKL oracle, thanks to a backtracking strategy that deals with the unavoidable errors in updating the belief region around the target. We evaluate the performance of the algorithm both over the posited oracle and over several real-world triplet datasets. We also report on a comprehensive user study, where human subjects navigate a database of face portraits.

## 1 INTRODUCTION

Searching a database in order to find a target item via some explicit query, such as one or several keywords, is a well-studied problem in information retrieval (IR). However, depending on the data type, it can be difficult or inefficient to formulate an explicit query. For example, the witness of a crime working with police does not sketch the face of a suspect; instead, she provides feedback on a sequence of images to gradually arrive at a faithful approximation of the

suspect's face. This is an example of *interactive comparison-based search*, where the user navigates towards the target item sequentially [Tschopp et al., 2011, Canal et al., 2019, Chumbalov et al., 2020]. In this approach, the user does not formulate an explicit query; rather, she answers a set of simple similarity queries with respect to the target: Among two items $i$ and $j$ provided by the system, which is closer to the intended target $t$? We refer to the outcome of such a query as a *triplet* $(i, j; t)$: among $\{i, j\}$, the user considered $i$ more similar to $t$.

The central component in such a system is a probabilistic oracle model that encapsulates how users answer such queries. Most approaches, including ours, posit that items live in some low-dimensional feature space. The embeddings of items $(\mathbf{x}_1, \mathbf{x}_2, \ldots, \mathbf{x}_n)$ in this feature space determine the noisy outcomes of triplet queries. Depending on the scenario, these embeddings can be derived from explicit item features (e.g., describing the geometry of a face), or they can be considered latent and estimated from past triplet data [Tamuz et al., 2011, Van Der Maaten and Weinberger, 2012, Chumbalov et al., 2020]. A *search algorithm* then presents a pair of items to the user, collects feedback, and repeats this process, until it can guess the target.

Chumbalov et al. [2020] posit a *Probit* oracle model that assumes that the probability of answering $i$ or $j$ depends on the distance of $t$ from the bisecting hyperplane between $\mathbf{x}_i$ and $\mathbf{x}_j$, relative to a noise parameter $\sigma_e$. They develop a search algorithm that maintains a Gaussian belief distribution over the embedding space, which captures the current knowledge about $\mathbf{x}_t$. Each query maximizes the information gain relative to this belief distribution, until the target is guessed correctly. A drawback inherent to their oracle model is that scaling to large $n$ is problematic due to the assumptions underlying the Probit oracle: once the belief distribution starts concentrating, the information contained in queries decreases. For example, suppose for exposition's sake that the algorithm has narrowed down the target to two candidates $t'$ and $t''$, and that the distance $\|\mathbf{x}_{t'} - \mathbf{x}_{t''}\|$ is small, relative to the noise parameter $\sigma_e$. Then *any possible*

---

*Authors contributed equally and are listed in alphabetical order.
†Work done while at EPFL.

*query pair* $(i, j)$ generates answers relative to $t'$ and to $t''$ that are nearly indistinguishable (i.e., they are Bernoulli random variables whose parameters are close). This means that the rate at which the belief distribution concentrates around $\mathbf{x}_t$ decreases, slowing down progress of the search. This leads to an unfavorable scaling of expected search cost when $n$ grows large[1].

In this paper, we argue that it is plausible to assume a more favorable oracle model. Specifically, we posit that the probability of choosing $i$ over $j$ is *scale-invariant* or *self-similar*, i.e., that it depends on the item embeddings only via the ratio of $\|\mathbf{x}_i - \mathbf{x}_t\|$ to $\|\mathbf{x}_j - \mathbf{x}_t\|$. In other words, to compare two very dissimilar items with respect to a target that is very dissimilar from both, is no harder (nor easier) than to compare two quite similar items to a nearby target. There is some evidence that this model reflects some of the psychological laws in perception [Chater and Brown, 1999, Laming, 1986], and we provide additional experimental evidence on this point in Section 4.

Under a perfectly scale-free oracle, the information required to halve the volume of the belief region does not depend on the scale of the current belief region. This suggests that there is hope that this volume can shrink exponentially fast with the number of queries. Indeed, a central contribution in this paper is an algorithm that achieves exponential convergence. In the noisy setting we study, this is non-trivial, because there is always the possibility of errors in oracle answers, such that the current belief moves too far away from the target. We solve this with a backtracking strategy that detects the occurrence of an error based on subsequent queries, and expands the belief region in order to "recapture" the target. We prove the exponential convergence of the expected distance to the target via an equivalence of a biased random walk on an infinite graph, which captures the containment relationships among the family of belief regions available to the algorithm.

**Related work.** A number of different triplet comparison models were introduced and studied in the machine-learning literature. Their main focus is on learning an embedding from comparison triplet data, which then allows predictions for unseen triplets. In Van Der Maaten and Weinberger [2012], the authors propose the t-STE model and capture the similarities between items via a Student-$t$ kernel, whose power-law tail confers robustness to outlier triplets. This model shares the drawback of the Probit model in that a narrow query ($\|\mathbf{x}_i - \mathbf{x}_j\| \to 0$) provides varnishing information, regardless of the target location. Later, Amid and Ukkonen [2015] generalize the idea of the t-STE model by allowing multiple representations of the same object in several different low-dimensional maps. The scale-invariant CKL model, introduced in Tamuz et al. [2011], corresponds

to the special case $\gamma = 2$ of the model considered in this paper. The Probit model is explored in Chumbalov et al. [2020] and Canal et al. [2019], where the output probability is a function of the distance between the target and the hyperplane bisecting the two query points. For a thorough discussion and comparison of (both noisy and noiseless) comparison triplet oracles and the embedding techniques they induce, see Vankadara et al. [2023].

A number of papers consider the problem of searching for a target using a sequence of *noiseless* comparison queries. Search via such comparison queries has been considered, for example, in Dasgupta [2005], Nowak [2008]. Karbasi et al. [2012] assume that all distances between pairs of items are known. Their analysis assumes either a noiseless oracle, or an oracle with constant error probability, independently of the distances between query items and target. This uniform noise model is not a realistic assumption for most applications, because it essentially assumes that every query conveys the same amount of information, independently of $\mathbf{x}_{i,j,t}$; if the true oracle is different, this assumption leads to inefficient search algorithms. Extensions of this line of work include oracles with ternary output including "I don't know" for similar query items [Kazemi et al., 2018], and larger query sets from which the most similar item is selected [Karbasi et al., 2015]. Although these approaches are similar in spirit to the problem considered here, the resulting search algorithms are not robust to noise, as they are unable to correct for incorrect query outcomes as the search progresses. Finally, there exists a line of work where noiseless triplet queries are used for efficient nearest-neighbor search in high-dimensional spaces [Haghiri et al., 2017].

The problem of searching in a space with *noisy* similarity queries is studied in Cox et al. [2000], Fang and Geman [2005], Ferecatu and Geman [2007], Suditu and Fleuret [2012], Garnett et al. [2012], using different comparison models in a fully Bayesian framework. In order to find the next query to ask, these methods usually aim to maximize the information gain by performing an exhaustive search over all combinations of pairs of items, which becomes prohibitively expensive for large $n$. This unfavorable computational efficiency was addressed in Canal et al. [2019] and Chumbalov et al. [2020], where the authors propose search schemes with more favorable tradeoffs between query and computational complexity by approximating the knowledge about $\mathbf{x}_t$ with a parametric distribution, which results in much better scalability. Comparison queries have also been explored in other active-learning scenarios, where, rather than finding one target item (or target point in a feature space), the goal is to determine a hypothesis function $h$ that assigns binary labels for all items in the database, assuming the two classes are separable by an unknown hyperplane [Kane et al., 2017, Nowak, 2009].

The remainder of this paper is structured as follows. In Section 2, we describe the $\gamma$-CKL model and explore the scaling

---

[1] We note that simply reducing $\sigma_e$ does not help, because this would make macroscopic queries too certain.

relationship between $\gamma$ and the embedding dimension $d$ with fixed error rate. In Section 3, we give the search algorithm for the dense case, i.e., when every point $\mathbf{x} \in \Omega \subset \mathbb{R}^d$ is a potential target. We formally prove that this algorithm shrinks the expected distance to the target exponentially fast. In Section 4, we compare $\gamma$-CKL against commonly used choice models on a series of comparison datasets and show the results of a comprehensive user study that validates the performance of the $\gamma$-CKL model. We also present synthetic experiments that confirm the exponential convergence rate of our new algorithm.

## 2 MODEL

For a query $Q = (\boldsymbol{x}_i, \boldsymbol{x}_j)$ and the corresponding oracle answer $Y$, call $p_{\boldsymbol{x}_i, \boldsymbol{x}_j, \boldsymbol{x}_t} = P(Y = \boldsymbol{x}_i \mid Q = (\boldsymbol{x}_i, \boldsymbol{x}_j), \boldsymbol{x}_t)$ the probability of the outcome $Y = \boldsymbol{x}_i$, i.e., "$i$ is closer than $j$ to $t$". In this paper, we discuss the advantages of a scale-free oracle model, for which $p_{c \cdot \boldsymbol{x}_i, c \cdot \boldsymbol{x}_j, c \cdot \boldsymbol{x}_t} = p_{\boldsymbol{x}_i, \boldsymbol{x}_j, \boldsymbol{x}_t}$ $\forall c \in (0, 1)$. To the best of our knowledge, among the models studied in the existing machine-learning literature, only Tamuz et al. [2011] have proposed such a scale-invariant choice model:

$$p_{\boldsymbol{x}_i, \boldsymbol{x}_j, \boldsymbol{x}_t}^{\text{CKL}} = \frac{||\boldsymbol{x}_j - \boldsymbol{x}_t||^2}{||\boldsymbol{x}_i - \boldsymbol{x}_t||^2 + ||\boldsymbol{x}_j - \boldsymbol{x}_t||^2}. \quad (1)$$

A shortcoming of (1) is its high sensitivity to the "curse of dimensionality": the probability of error grows quickly with $d$. Indeed, for a fixed target point and two query points sampled uniformly at random from a ball around the target, the predicted probability of the closest point to be chosen by the oracle (1) decays to 1/2 for $d \to \infty$. This makes comparison-based searching difficult, because most queries would provide almost no information about the target's location. We propose a simple generalization of (1) that addresses this shortcoming:

$$p_{\boldsymbol{x}_i, \boldsymbol{x}_j, \boldsymbol{x}_t} = \frac{||\boldsymbol{x}_j - \boldsymbol{x}_t||^\gamma}{||\boldsymbol{x}_i - \boldsymbol{x}_t||^\gamma + ||\boldsymbol{x}_j - \boldsymbol{x}_t||^\gamma}, \quad (2)$$

with $\gamma > 0$. The parameter $\gamma$ controls the power of the oracle independently of the embedding dimension $d$. To see this, note that when $\gamma$ is fixed and $d \to \infty$, the probability (2) for a uniformly selected pair of points $\boldsymbol{x}_i, \boldsymbol{x}_j$ goes to 1/2. On the other hand, when $\gamma \to \infty$ and $d$ is fixed, this probability becomes an indicator function $p_{\boldsymbol{x}_i, \boldsymbol{x}_j, \boldsymbol{x}_t} \to \mathcal{I}\{||\boldsymbol{x}_i - \boldsymbol{x}_t|| < ||\boldsymbol{x}_j - \boldsymbol{x}_t||\}$. This suggests that as the dimension $d$ of the space grows, the new model should enable us to control the average outcome bias by scaling the parameter $\gamma$ accordingly. In the following theorem, we show that a linear scaling relationship between $\gamma$ and $d$ achieves this:

**Theorem 2.1.** *Consider a $d$-dimensional ball $\mathcal{B} \subset \mathbb{R}^d$ of radius 1. Let the target point $\mathbf{x}_t$ be the center of $\mathcal{B}$. For two points $\boldsymbol{x}_a, \boldsymbol{x}_b$ sampled uniformly from $\mathcal{B}$, let $p_Q$ be the probability of the correct answer on a query $Q = (\boldsymbol{x}_a, \boldsymbol{x}_b)$ given the target $\mathbf{x}_t$. For any $c_2 \in [\frac{1}{2}, 1]$ there is a constant $c_1 > 0$ such that if $\gamma$ grows linearly with $d$, $\gamma = c_1 d + o(d)$, then $p_Q \to c_2$.*

We provide some intuition on a condition for the geometric structure of a set of queries to be rich enough to identify the target $\boldsymbol{x}_t \in \mathbb{R}^d$ under the $\gamma$-CKL model. In particular, the following proposition (proven in the appendix) establishes an identifiability condition of the target $\boldsymbol{x}_t$ for a finite set of queries $\hat{\mathcal{Q}} = \{\hat{Q}_1, \hat{Q}_2, \ldots, \hat{Q}_L\}$ for which the exact answer probabilities are known (or alternatively, that are each repeated infinitely many times so that the answer probabilities can be exactly estimated). Each query constrains the locus of $\boldsymbol{x}_t$ to a $d - 1$-dimensional sphere; if these spheres intersect in only one point, it is at $\boldsymbol{x}_t$ (cf. Fig. 1).

**Proposition 2.2.** *Assume that $\Omega \subset \mathbb{R}^d$ is $d$-dimensional compact set and that the target $\boldsymbol{x}_t$ is sampled uniformly at random from $\Omega$. Consider an infinite sequence of queries $\mathcal{Q} = \{Q_0, Q_1, \ldots\}$ that is asked to the oracle, where each $Q_i \in \hat{\mathcal{Q}} = \{\hat{Q}_1, \hat{Q}_2, \ldots, \hat{Q}_L\}$ and each $\hat{Q}_i$, $i = 1, 2, \ldots, L$, is repeated infinitely many times. Also for each $\hat{Q}_i = (\hat{\boldsymbol{x}}_i^a, \hat{\boldsymbol{x}}_i^b)$ let $c_i = ||\hat{\boldsymbol{x}}_i^a - \boldsymbol{x}_t||/||\hat{\boldsymbol{x}}_i^b - \boldsymbol{x}_t||$ and $\hat{\boldsymbol{z}}_i = (c_i \hat{\boldsymbol{x}}_i^b - \hat{\boldsymbol{x}}_i^a)/(1 - c_i)$. If $\hat{\mathcal{Q}}$ satisfies $\text{rank}(\boldsymbol{Z}) = d$, where $\boldsymbol{Z}$ is the $d \times (L-1)$ matrix of vectors $\{(\hat{\boldsymbol{z}}_i - \hat{\boldsymbol{z}}_L) : \hat{Q}_i \in \hat{\mathcal{Q}}, i = 1, \ldots, L-1\}$, then $\arg\max_{\boldsymbol{x} \in \Omega} \mathbb{E}[p(\boldsymbol{x} \mid Y_{1:m})] \to \boldsymbol{x}_t$ as $m \to \infty$.*

A natural question to ask is: Is the scale-free model we propose a reasonable proxy for human comparisons? In Section 4.1, we study this question empirically by using several real-world datasets. We learn latent embeddings by maximizing the product of likelihoods (2) on a training set, and evaluate the accuracy on a hold-out set. We answer the question in the affirmative, and find that the addition of the $\gamma$ parameter enables our model to perform favorably when compared to other commonly used choice models.

In the next section, we first focus on the search problem, and assume that item embeddings are known.

## 3 ADAPTIVE SEARCH ALGORITHM

We now consider a scenario where the search space $\Omega$ is a hypercube in $\mathbb{R}^d$ and where any $\mathbf{x}_t \in \Omega$ can be the target. In this continuous setting, a search algorithm should be able to "zoom in" indefinitely, thus finding ever smaller regions containing the target.

Having access to a scale-free oracle enables us to ask queries where the response noise is independent of the current scale (or "zoom level"). A constant level of noise in the oracle's answers means that shrinking the current region incurs constant expected cost in terms of the number of queries asked. This suggests an exponential rate of convergence, as long

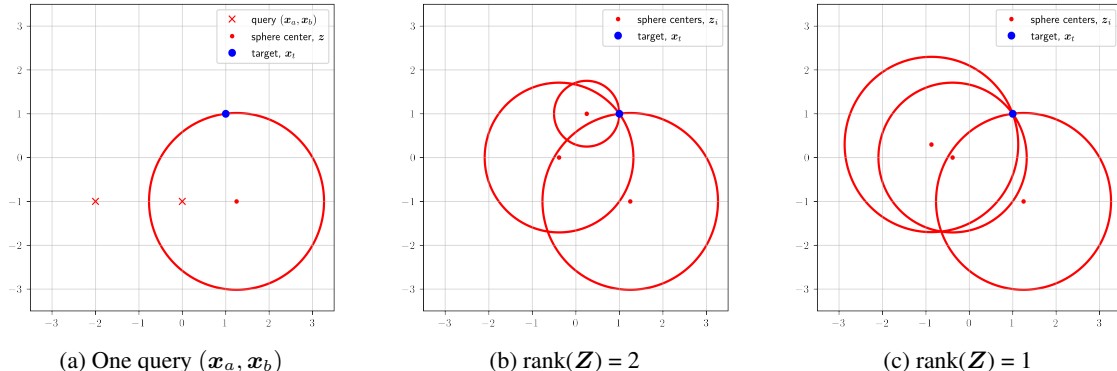

|                   |                   |                   |
|:-----------------:|:-----------------:|:-----------------:|
| (a) One query $(\boldsymbol{x}_a, \boldsymbol{x}_b)$ | (b) rank$(\boldsymbol{Z}) = 2$ | (c) rank$(\boldsymbol{Z}) = 1$ |

Figure 1: Illustration of the result of Proposition 2.2 in $\mathbb{R}^2$. (a) For each query the subset of points in $\Omega$ that maximizes the expected log-likelihood geometrically is a sphere containing $\boldsymbol{x}_t$ with center $\boldsymbol{z}$. (b) When the set of sphere centers $\{\boldsymbol{z}_i\}$ corresponding to the queries $\hat{\mathcal{Q}}$ span a volume in $\mathbb{R}^2$, these spheres intersect at exactly one point, $\boldsymbol{x}_t$. (c) Otherwise, there are multiple points of intersection, and $\boldsymbol{x}_t$ is not identifiable.

as we can ensure that the search does not permanently veer away from the target. For the formal analysis, we assume that items are dense within the feature space, so that the target and query items may be at any location. This allows us to reason about the speed of convergence of the search process towards the target, instead of a stopping time of that process over a finite set of items. In real-world scenarios with a finite number of items, we believe the theory gives us the following insight. Starting with a large number of items, we expect the situation to be similar to the dense case we study theoretically, and informally we expect the algorithm to be able to "zoom in" on the target with an exponential rate of convergence, until it has arrived at a zoom level at which the dataset begins to look sparse. At that point, the theory is no longer applicable, but we expect to have filtered the search space down to a small number of items, such that identifying the target object among the remaining items is much easier to do.

Our algorithm operates in stages. At each stage, the algorithm investigates a region by submitting queries until a decision to zoom in or backtrack can be made. This decision is based only on information collected in the current stage. At the beginning of a stage, all knowledge about prior queries and oracle replies is discarded, and the only state of the algorithm is the current region. Due to this conditional independence of decisions, we find that the sequence of regions visited by our algorithm is Markovian. We frame our search process as a random walk on a graph, where each node corresponds to a region $X \in \Omega$. Under mild assumptions on the transition probabilities between regions, an erroneous decision, i.e., zooming into a region that does not contain the target, must eventually be undone with probability 1. A high-level overview of this idea is given in Algorithm 1. Constructing a stochastic coupling between a counting process and the random walk on regions enables us to analyze the probability of consecutive errors and to ex-

---

**Algorithm 1** Exponential search algorithm

**Input:** query budget $M$
$s \leftarrow 0$ {stage number}
$X_s \leftarrow \Omega$ {current region}
$m \leftarrow 0$ {number of queries asked}
**repeat**
  $\mathcal{D} \leftarrow \{\}$ {Drop previous observations}
  **repeat**
    $(\hat{\mathbf{x}}_i, \hat{\mathbf{x}}_j) \leftarrow \text{nextQuery}(X_s, \mathcal{D})$
    $\hat{y} \sim \text{Bernoulli}(p_{\hat{\mathbf{x}}_i, \hat{\mathbf{x}}_j, \mathbf{x}_t})$
    $\mathcal{D} \leftarrow \mathcal{D} \cup \{(\hat{\mathbf{x}}_i, \hat{\mathbf{x}}_j, \hat{y})\}$
    $m \leftarrow m + 1$
  **until** decisionReady$(X_s, \mathcal{D}) = \text{true}$
  $X_{s+1} \leftarrow$ zoom into / out of $X_s$ based on $\mathcal{D}$
  $s \leftarrow s + 1$
**until** $m > M$

---

plicitly calculate recurrence times. We prove an exponential rate of convergence in Section 3.1.

We show that with access to a $\gamma$-CKL model, the assumptions on transition probabilities can always be satisfied. In Section 3.2, to ensure only a constant number of queries is needed in each stage, we present a query scheme that relies on the properties of a scale-free oracle. To facilitate a formal proof, this scheme is based on hypothesis testing. We discuss an efficient implementation based on numerical integration in Section 3.3.

### 3.1 CONVERGENCE ANALYSIS

Let $X$ be the current belief region of the search process. We assume that $X$ is a unit hypercube centered at the origin. Here, we constrain our analysis to hypercubes, nevertheless, the idea of the algorithm applies to arbitrary regions. Let $S$

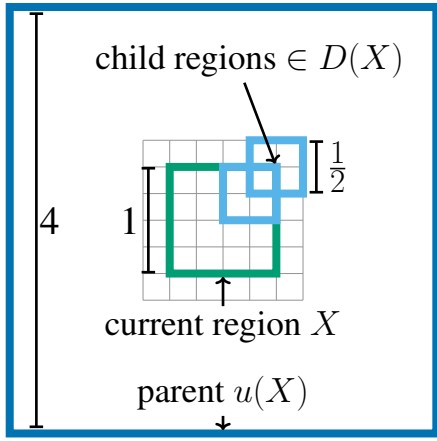

Figure 2: $X$, two children and parent region.

be a hypercube centered at the origin with edge length $\frac{3}{2}$. Let $\mathcal{T}_{S,\frac{1}{4}}$ be a set of hypercubes with edge length $\frac{1}{4}$ that tile $S$. We define the set of *children* $D(X)$. $D(X)$ is the set of all hypercubes of edge length $\frac{1}{2}$ that can be constructed by joining tiles in $\mathcal{T}_{S,\frac{1}{4}}$. Figure 2 illustrates this construction. We see that along each axis, there are five possible positions for a child, which give us a total of $5^d$ children. The hypercube of edge length $4$, centered at the origin contains all regions $X'$ for which $X \in D(X')$. It is the union of all direct ancestors of $X$, for convenience we will refer to it as the *parent* $u(X)$ of $X$. In our algorithm, backtracking from $X$ leads to $u(X)$.

If $\mathbf{x}_t \in X$ then we call $X$ green (correct), otherwise red (incorrect). A green region must have at least one green child. The parent of a green region must be green.

At every stage, our algorithm collects query replies, until it makes a decision to proceed to one of the child regions or to backtrack to the parent. Similarly to the classification of regions, we distinguish between correct and incorrect decisions. Proceeding to a green node is correct, whereas backtracking from a green node is incorrect. Proceeding to a red node is incorrect, whereas backtracking from a red node is correct. The probability of all these events depends on the current region $X$ and the target $\mathbf{x}_t$. Table 1 shows a comprehensive listing. We refer to the probabilities associated with a correct transition with $p$ and incorrect transitions with $q$. For a green region, there are two incorrect decisions: backtracking and straying by proceeding to a red child. They are named $q_u$ and $q_s$. The correct decision is to proceed to a green child, it is named $p_d$. For a red region, there are two correct decisions: backtracking and recovering by proceeding to a green child. They are named $p_u$ and $p_r$. The incorrect decision is to proceed to a red child, it is named $q_d$. For the analysis of a random walk on colored regions we need the total probability of a correct or incorrect decision. This is also shown in Table 1 An illustration with nested regions and the corresponding transitions is shown in Figure

Table 1: Transition probabilities.

| $X$ IS GREEN: | |
| --- | --- |
| TRANSITION TO | PROBABILITY |
| PARENT | $q_u(X, \mathbf{x}_t)$ |
| GREEN CHILD | $p_d(X, \mathbf{x}_t)$ |
| RED CHILD | $q_s(X, \mathbf{x}_t)$ |
| CORRECT | $p(X, \mathbf{x}_t) = p_d(X, \mathbf{x}_t)$ |
| INCORRECT | $q(X, \mathbf{x}_t) = q_u(X, \mathbf{x}_t) + q_s(X, \mathbf{x}_t)$ |

| $X$ IS RED: | |
| --- | --- |
| TRANSITION TO | PROBABILITY |
| PARENT | $p_u(X, \mathbf{x}_t)$ |
| RED CHILD | $q_d(X, \mathbf{x}_t)$ |
| GREEN CHILD | $p_r(X, \mathbf{x}_t)$ |
| CORRECT | $p(X, \mathbf{x}_t) = p_u(X, \mathbf{x}_t) + p_r(X, \mathbf{x}_t)$ |
| INCORRECT | $q(X, \mathbf{x}_t) = q_d(X, \mathbf{x}_t)$ |

3.

**Lemma 3.1.** *The sequence of regions $X_s$ visited in each stage $s$ of the search process forms a random walk.*

Intuitively, we need the probability of making a correct decision to be strictly higher than the probability of an incorrect decision. This is formalized in the following definition: Let $b > 0$ be a constant, such that for any $X \subset \Omega$ that can be visited by our algorithm, and any $\mathbf{x}_t$:

**Assumption 3.2.** $p(X, \mathbf{x}_t) - q(X, \mathbf{x}_t) > b$.

**Assumption 3.3.** $\mathbf{x}_t \in X \implies p_d(X, \mathbf{x}_t) - 2q_u(X, \mathbf{x}_t) - q_s(X, \mathbf{x}_t)\frac{b+1}{2b} > 0$.

Assumption 3.3 is designed to facilitate the proof of Theorem 3.8.

In practice, it is simple to tune the confidence with which the search makes its decisions: Collecting more queries before committing to a decision decreases the chance of making a mistake. The next theorem asserts that, with access to a scale-free oracle, it is always possible to satisfy Assumptions 3.2 and 3.3. We constructively prove Theorem 3.4 by presenting Algorithm 2 in Section 3.2.

**Theorem 3.4.** *For any $b$ and any $X$, there is an algorithm that needs to observe, at most, a constant and finite number of replies from a $\gamma$-CKL oracle, until it can make a decision with probabilities that satisfy Assumptions 3.2 and 3.3.*

We keep track of the number of incorrect decisions. Let $z(\mathbf{x}_t, X)$ be the number of backtracking decisions that are needed to reach a green region from $X$. If the search proceeds to a red child, $z(\mathbf{x}_t, X)$ is either increased by 1, or

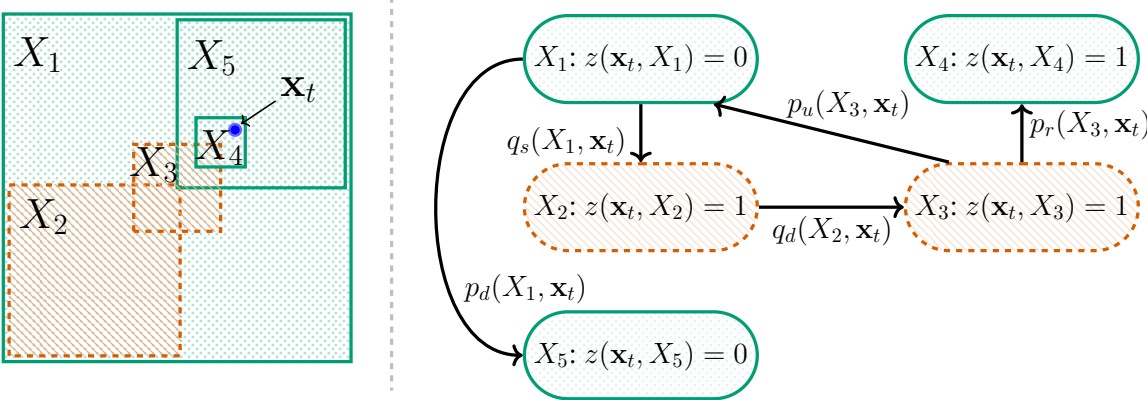

Figure 3: Regions and target (blue dot) on the left side, selected transitions on the right side.

stays unchanged (it is possible that no additional backtracking is required). Recovering, by proceeding to a green child, means immediately setting $z(\mathbf{x}_t, X)$ to 0.

From Assumption 3.2 we get $\mathbf{x}_t \in X \implies q_u(X, \mathbf{x}_t) + q_s(X, \mathbf{x}_t) < \frac{1-b}{2}$ and $\mathbf{x}_t \notin X \implies q_d(X, \mathbf{x}_t) < \frac{1-b}{2}$. We construct a time-homogenous random walk that will serve as a stochastic upper bound for $z(\mathbf{x}_t, X_s)$. Let $Z_s$ be a random walk on natural numbers, starting at $Z_0 = z(\mathbf{x}_t, X_0)$. At each step, $Z_s$ is incremented with probability $\frac{1-b}{2}$ and decremented with probability $\frac{1+b}{2}$. Once $Z_s$ reaches 0, there is a self loop of probability $\frac{1+b}{2}$ and a transition to 1 with probability $\frac{1-b}{2}$.

**Lemma 3.5.** *Given a stochastic decision criterion that satisfies Assumption 3.2, $Z_s$ is a stochastic upper bound for $z(\mathbf{x}_t, X_s)$, we denote this by $z(\mathbf{x}_t, X_s) \preceq_{st.} Z_s$*

*Proof sketch.* We construct a coupling between the random walk $\tilde{X}_s$ and a random variable $\tilde{Z}$. We then use induction to show that with probability 1 it holds that $\tilde{Z} > z(\mathbf{x}_t, \tilde{X}_s)$. □

**Lemma 3.6.** *Given a stochastic decision criterion that satisfies Assumption 3.2, for any $k > 0$*

$$\mathbf{P}[z(\mathbf{x}_t, X_s) > k] \leq \left(\frac{1-b}{1+b}\right)^k .$$

Let $\tau_X = \inf\{s > 0 \mid \mathbf{x}_t \in X_s, X_0 = X\}$ be the stopping time of reaching a green region, starting from $X$.

**Lemma 3.7.** *. Let $X$ be red and $u(X)$ be green (this occurs after just having strayed from a green region). Given a stochastic decision criterion that satisfies Assumption 3.2, it holds that $\mathbb{E}[\tau_X] \leq \frac{1}{b}$.*

*Proof sketch.* The proof relies on $Z_s$ as a stochastic upper bound. We first use the Ergodic Theorem to prove the existence of a unique stationary distribution. We then explicitly

calculate this stationary distribution and use it to derive recurrence times. This enables us to prove an upper bound on the expected stopping time. □

To quantify the progress of our search, we keep track of the *depth* $P(X)$ of a region. The depth is the number of consecutive proceed decisions needed to reach this region, starting from $\Omega$. The edge length of a region $X$ at depth $P(X)$ is $\left(\frac{1}{2}\right)^{P(X)}$. The $k$-th ancestor $u(X, k)$ is reached by backtracking $k$ times from $X$. With the following theorem we show the exponential rate of convergence of our algorithm. At every stage $s$ of the algorithm, $u(X, k)$ contains the target with high probability (which doesn't depend on $s$) and its depth increases at a linear rate.

**Theorem 3.8.** *Given a subroutine that satisfies Assumptions 3.2 and 3.3, for any desired probability of error $\delta$, there are two constants $k > 0$ and $C > 0$ such that*

$$\mathbf{P}\left[\mathbf{x}_t \in u(X_s, k)\right] > 1 - \delta, \quad \mathbf{E}\left[P(u(X_s, k))\right] > Cs.$$

*Proof sketch.* We define a stopping time of arriving at a green region after leaving a green region. Using the results of Lemma 3.7, we prove an upper bound for the expectation of this stopping time. Using Assumption 3.3, we show that the expected depth of each consecutive green region increases linearly. Together with Lemma 3.6, the statement follows. □

### 3.2 A SCALE-FREE DECISION CRITERION

In each stage, we need a querying scheme that asks at most a constant number of queries, until it arrives at a decision. The probability of error needs to satisfy Assumptions 3.2 and 3.3.

Our scheme is based on a test for the hypothesis (H) "$\mathbf{x}_t$ *is in the region* $X$". As $\mathbf{x}_t$ approaches the boundary of $X$, it becomes increasingly hard to distinguish whether the point

is inside or outside. This leads to a region of uncertainty $U$ around $X$ in which our hypothesis test is not reliable.

Let $X$ be a hypercube of edge length 2, centered at the origin and let $U$ be a hypersphere with radius $r_u > 1$, also centered at the origin. Everything outside of $U$ is $F = \Omega \setminus U$. We will construct a query $Q$ and calculate the corresponding $r_u$ such that repeatedly observing the outcome of $Q$ enables us, with probability $> 1 - \delta$, to accept (H) if $\mathbf{x}_t \in X$, or to reject (H) if $\mathbf{x}_t \in F$.

**Lemma 3.9.** *We assume $d > 1$. Let $Q = (0, (1+d)\mathbf{e})$, where $\mathbf{e} = (1, 0, 0, \dots)$ is a unit vector along an arbitrarily chosen axis. Let $r_u = 1 + \frac{d + \sqrt{d^3 + d^2} - d}{d - 1}$. Let $X, U, F$ be defined as above. Then for any delta $\delta > 0$ observing a constant number of query outcomes is enough to apply a one-tailed binomial hypothesis test which will with probability $1 - \delta$: accept (H), if $\mathbf{x}_t \in X$, or reject (H), if $\mathbf{x}_t \in F$. The necessary number of observations does not depend on $X$ and $\mathbf{x}_t$.*

We need to find out whether a child of the current belief region contains the target. Due to the region of uncertainty, we cannot apply the hypothesis test directly to the child regions. Instead, we construct a finer discretization grid.

In Lemma 3.9, we assume a region of edge length 2. As our oracle model is scale-free, we can apply the hypothesis test to a smaller region, which results in a smaller uncertainty region as well. For a region with edge length $r_c$, the radius of the uncertain region is scaled by $r_c/2$. Let $r_c < \frac{1}{8r_u}$, which leads to an uncertain region with radius $r_u \frac{r_c}{2} < \frac{1}{16}$.

Let $\mathcal{T}(S, r_c,)$ be a tiling of $S$ with hypercubes of edge length $r_c$, we refer to the cells in this tiling by $c_k, k = 1..K$, the respective centers are $x_{c_k}$. If the edge length of $S$ is not divisible by $r_c$, it is always possible to pick a smaller value for $r_c$. Each cell $c_k$ in the tiling belongs to one of these classes:

- (A) $\mathbf{x}_t \in c_k$. When using the hypothesis test, with high probability, our test will not reject (H). We assume that cells include their border. If the target happens to lie exactly on the boundary between cells, then all of them belong to class (A).
- (B) $\mathbf{x}_t \notin c_k \wedge \|\mathbf{x}_t - \mathbf{x}_{c_k}\| < \frac{1}{16}$. When using the hypothesis test, the target lies in the uncertain region. We do not make any assumption about whether (H) is rejected or not.
- (C) $\mathbf{x}_t \notin c_k \wedge \|\mathbf{x}_t - \mathbf{x}_{c_k}\| \geq \frac{1}{16}$. When using the hypothesis test, with high probability, our test will reject hypothesis (H).

When using the hypothesis test, we know that, with high probability, all cells in class (C) are rejected. The remaining cells fit in a small bounding box. This is illustrated in Figure 4.

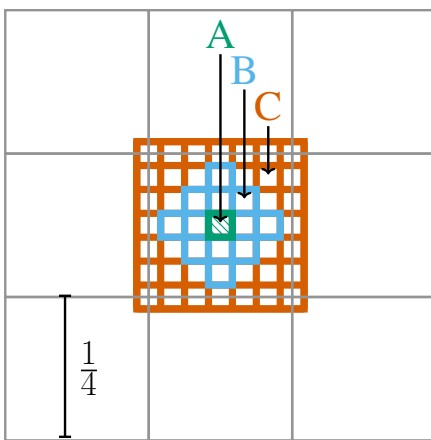

Figure 4: Layout of the nested grid. The target lies in cell A.

---

**Algorithm 2** Search with hypothesis test criterion

Set up a discretization $\mathcal{T}(S, r_c,)$.
$H \leftarrow \emptyset$

This loop replaces the nextQuery subroutine from Algorithm 1
**for** $c_k$ in $\mathcal{T}(S, r_c,)$ **do**
    perform the hyp. test from Lemma 3.9 for $c_k$
    **if** hypothesis is not rejected **then**
        $H \leftarrow H \cup c_k$

This criterion corresponds to decisionReady from Algorithm 1
**if** $\exists \hat{X} \in D(X_s) : H \subseteq \hat{X}$ **then**
    proceed to $\hat{X}$
**else**
    backtrack to parent $u(X_s)$

---

**Lemma 3.10.** *There is a hypercube $\mathcal{B}$ with an edge length of less than $\frac{1}{4}$, such that all cells in the classes (A) and (B) are fully contained in $\mathcal{B}$.*

We apply the hypothesis test to all cells in $\mathcal{T}(S, r_c,)$. Let $\mathcal{B}$ be the bounding box containing all cells for which (H) was not rejected. If $\mathcal{B}$ does not overlap with $X$, we backtrack. Otherwise, if there is a child region that fully contains $\mathcal{B}$, we proceed to it. This mechanism is formalized in Algorithm 2. The following Theorem 3.11 shows that this algorithm enables us to make decisions that lead to an exponential rate of convergence, i.e., they satisfy Assumptions 3.2 and 3.3.

**Theorem 3.11.** *Algorithm 2 can achieve any desired probability of error $\hat{\delta}$, while requiring only a finite number of queries. In particular, choosing $\hat{\delta}$ small enough ensures that the scheme is compatible with Assumptions 3.2 and 3.3.*

### 3.3 IMPLEMENTATION

In practice, it is not efficient to conduct a series of independent hypothesis tests. A real-world implementation should, instead, rely on numerical integration. Within each stage, the algorithm collects oracle replies and updates an approximation of the posterior distribution of the target location, until a decision can be made. The *nextQuery* subroutine in Algorithm 1 then corresponds to a random sample based on the current belief region. As more and more evidence from queries is collected, the posterior distribution will either concentrate in a child region, or show that the target is likely not in the current belief region. We can define a confidence threshold $\alpha$: The algorithm proceeds if there is a subregion $\hat{X}$ with $\int_{\hat{X}} p(\mathbf{x}_t|Q, y) > \alpha$ and backtracks if $\int_X p(\mathbf{x}_t|Q, y) < 1 - \alpha$, this prescribes a criterion for *decisionReady* in Algorithm 1. Please refer to the supplementary material for an implementation in Python and to Section 4.3 for a benchmark of our algorithm based on synthetic data.

## 4 EMPIRICAL EVALUATION

### 4.1 OFFLINE COMPARISONS DATASETS

In order to validate how well our proposed model fits the real world data, we compare it with the oracle choice models from the literature, namely t-STE, CKL, and Probit in an experiment on three real-world triplet comparisons datasets: Musical Artists Ellis et al. [2002] containing 9'107 triplets of $n = 400$ musicians, Food Wilber et al. [2014] containing 190'376 triplets of $n = 100$ food images, and Movie Actors Chumbalov et al. [2020] containing 50'026 triplets of $n = 552$ actors. For each dataset, we learned an embedding for every model using a different number of dimensions $d$ and performed a 10-fold cross-validation to compute the resulting accuracy on a holdout set. The hyperparameters for each model were optimized and the best performing configurations for each $d$ are reported. Overall, across the three datasets $\gamma$-CKL correctly predicts between 84% and 86% of triplets, see Fig. 5. For Musical Artists $\gamma$-CKL is on par with t-STE and outperforms Probit and CKL. For Food, $\gamma$-CKL are on par with t-STE and Probit and significantly outperforms CKL.. For Movie Actors dataset, $\gamma$-CKL outperforms its competitors. We can see that the $\gamma$-CKL model immediately benefits from having a general $\gamma$ parameter already in small dimensions compared to the original CKL. We can conclude that the new proposed oracle model very well reflects the real user behaviour on the comparison-like tasks. We also note that increasing $d$ benefits the quality of the learned embedding for $\gamma$-CKL, and as $d$ increases, the best performing values of $\gamma$ tend to also increase, which is aligned with the findings of Theorem 2.1 (see Appendix).

### 4.2 INTERACTIVE USER-STUDY

We are interested in the performance of a scale-free oracle model for the purpose of interactive search. The current state of the art is GAUSSSEARCH, as benchmarked in a user study by Chumbalov et al. [2020]. To compare $\gamma$-CKL to the Probit model underpinning GAUSSSEARCH, we implement an algorithm $\gamma$-CKLSEARCH similar in the spirit to GAUSSSEARCH, based on the likelihood predicted by $\gamma$-CKL (see Appendix). We then compare the two algorithms in a user study designed to mimic the setting of Chumbalov et al. [2020]. We not only find that GAUSSSEARCH performs slightly better than in the original study (thus validating the state-of-the-art), but also observe a significantly better search performance with $\gamma$-CKL.

Our set of items contains $n = 513$ pictures of famous movie actors[2]. At each step of a search, the user is presented with four pictures of faces of yet unseen actors and is asked to choose the one that resembles her target the most. The search is complete once the user finds her target, i.e., when the picture of the target's face appears in one of the four displayed pictures. An embedding of actors' faces has been learned individually for each algorithm, from triplets collected prior to the experiment.

Our study is designed with controlled randomization. Each user sees a target at most once. Each target is searched for twice, once with algorithm $\gamma$-CKLSEARCH and once with GAUSSSEARCH. This corresponds to an across-subject design and reduces item-related bias. To reduce user-related bias, we also use a within-subject design, where each user performs the same amount of searches with each of the two algorithms. The order in which searches are seen is random. Users are not aware of the algorithm they are testing. In total, we recruited 24 participants. We collected 207 search trajectories, 104 with GAUSSSEARCH and 103 with algorithm $\gamma$-CKLSEARCH. Our new method outperforms GAUSSSEARCH: with $\gamma$-CKLSEARCH a user needs on average **18.83±1.257** queries to find the target, whereas with GAUSSSEARCH he needs on average **22.08 ± 1.658** queries. $\gamma$-CKLSEARCH algorithm tends to ask queries that are cognitively easier for humans to answer: on average participants were spending 11.62 seconds to decide on a query during a search with $\gamma$-CKLSEARCH versus 13.19 seconds for a query from GAUSSSEARCH.

### 4.3 SYNTHETIC DATA

We created an open-sourced version of our algorithm, based on PyTorch Paszke et al. [2019] and provide it in the supplementary material. A synthetic evaluation of our search algorithm is shown in Figure 6. To illustrate the robustness of our algorithm, we show a variety of constellations of

---

[2]A demo version of this experiment is available under `https://who-is-th.at`

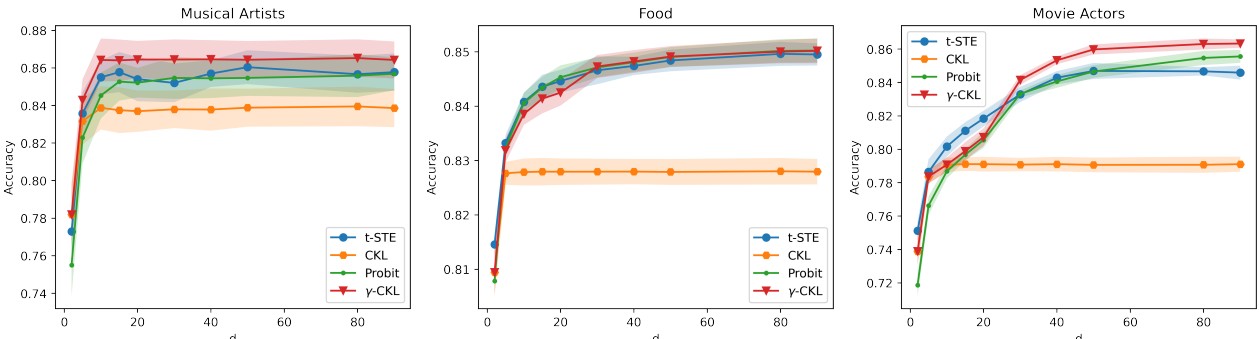

Figure 5: Quality of the embedding produced using different comparison models. Accuracy is reported on a 10-fold holdout triplet set. Our $\gamma$-CKL models either beat or are on par with competing oracle models.

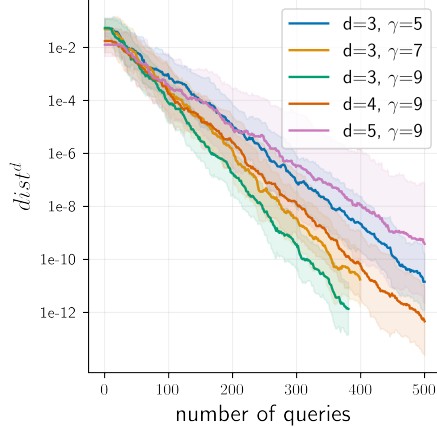

Figure 6: Exponential convergence across a range of dimensions and values for $\gamma$.

$\gamma$ and $d$; for each, we use 50 individual runs to compute confidence intervals. Our implementation is Algorithm 1 based on the heuristic from Section 3.3. The volume of a belief area scales with $O(d)$, to be able to compare the convergence rate across different values for $d$, we present the distance to the target, to the power of $d$. The implementation as well as additional visualizations are included in the supplementary material.

## 5 CONCLUSION

We have introduced $\gamma$-CKL, a scale-free oracle model with a parameter $\gamma$ to control the power of the oracle. $\gamma$-CKL yields embeddings that are competitive with commonly used choice models. It scales favourably to high embedding dimensions, a key improvement over CKL.

In the context of interactive search, $\gamma$-CKL outperforms a state-of-the-art implementation based on the Probit model. Our user study reproduces the existing results in a blind randomized trial and establishes statistically significant improvements for $\gamma$-CKL over *Probit*. Interestingly, we observed not only a reduction in the number of search steps on average, but also a reduction in the average time our subjects took to answer queries. This suggests a lower cognitive overhead, and provides further evidence that $\gamma$-CKL is well suited to model choices made by a human oracle.

At the same time, $\gamma$-CKL is only one representative of a large family of scale-free oracle models. Any model that bases its decision only on the ratio of distances between the query points and the target shows scale-free properties. We hope that our results motivate future researchers to study this type of oracle model.

In particular, a scale-free model enables a new class of efficient search schemes. Framing the search process as a random walk enables us to construct a scheme with provably exponential convergence. We believe that the algorithm discussed in Section 3 would be particularly suitable for searching for procedurally generated content. In such a setting, viable parametrizations are often continuous. In our work, we have focused on a rigorous proof of the exponential convergence rate. Future researchers can build on our insights by improving the query efficiency of the algorithm. Introducing active learning to the stages of the exponential algorithm, for example by optimizing expected information gain, is a promising extension.

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

# Fast Interactive Search under a Scale-Free Comparison Oracle
## (Supplementary Material)

**Daniyar Chumbalov**[‡ §1]    **Lars Klein**[*1]    **Lucas Maystre**[2]    **Matthias Grossglauser**[1]

[1]INDY Lab, EPFL, Lausanne, Switzerland
[2]Spotify, London, UK

## A TRIPLET OUTCOME PREDICTION EXPERIMENT DETAILS

Below we present details on the experiments from Section 4.1.

The hyperparameters for each model were optimized and the best performing configurations for each $d$ are reported. We used the following grid of hyperparameters: $lr \in [1e-2, 1e-3, 1e-4, 1e-5]$, $batchsize \in [128, 256, 512, |\mathcal{T}|]$, $L_2$ regularizer $\lambda \in [0, 0.4, 1]$, $\sigma_\varepsilon \in [0.4, 0.3, 0.2, 0.1, 0.01]$, $\gamma \in [2, 3, 5, 10, 15, 20, 25]$. No regularization was used for CKL and $\gamma$-CKL because these models are scale-free.

The best hyperparemeter configuration for each dataset are given below:

- **Musical Artists**
    - t-STE (accuracy 86%, nll 0.333): $D = 50$, $lr = 1e-4$, $\lambda = 0$, $batchsize = |\mathcal{T}|$
    - CKL (accuracy 83.9%, nll 0.395): $D = 80$, $lr = 1e-5$, $batchsize = 256$
    - Probit (accuracy 85.6%, nll 0.352): $D = 90$, $lr = 1e-2$, $\lambda = 0.4$, $\sigma_\varepsilon = 0.1$, $batchsize = 512$
    - $\gamma$-CKL (accuracy 86.5%, nll 0.329): $D = 80$, $lr = 1e-3$, $\gamma = 5$, $batchsize = |\mathcal{T}|$
- **Food**
    - t-STE (accuracy 84.9%, nll 0.327): $D = 80$, $lr = 1e-2$, $\lambda = 0$, $batchsize = |\mathcal{T}|$
    - CKL (accuracy 82.8%, nll 0.389): $D = 80$, $lr = 1e-3$, $batchsize = |\mathcal{T}|$
    - Probit (accuracy 85%, nll 0.327): $D = 90$, $lr = 1e-2$, $\lambda = 1.0$, $\sigma_\varepsilon = 0.01$, $batchsize = |\mathcal{T}|$
    - $\gamma$-CKL (accuracy 85.%, nll 0.327): $D = 90$, $lr = 1e-4$, $\gamma = 25$, $batchsize = |\mathcal{T}|$
- **Movie Actors**
    - t-STE (accuracy 84.6%, nll 0.4): $D = 50$, $lr = 1e-2$, $\lambda = 0$, $batchsize = 512$
    - CKL (accuracy 79.1%, nll 0.452): $D = 15$, $lr = 1e-2$, $batchsize = 512$
    - Probit (accuracy 85%, nll 0.327): $D = 90$, $lr = 1e-4$, $\lambda = 0,$, $\sigma_\varepsilon = 0.2$, $batchsize = |\mathcal{T}|$
    - $\gamma$-CKL (accuracy 86.3%, nll 0.314): $D = 90$, $lr = 1e-3$, $\gamma = 20$, $batchsize = |\mathcal{T}|$

## B $\gamma$-CKLSEARCH FOR MODERATE $n$

In the case when there is only a finite number $n$ of points, we can keep the full posterior distribution $\mathcal{P} = [p_1, p_2, \ldots, p_n]$ over all $n$ objects and propose a more efficient algorithm that the ones introduced in the previous subsection for continuous $\Omega$.

---

[*]Authors contributed equally and are listed in alphabetical order.
[†]Work done while at EPFL.
[‡]Authors contributed equally and are listed in alphabetical order.
[§]Work done while at EPFL.

---

**Algorithm 3** $\gamma$-CKLSEARCH

---

1: $m \leftarrow 0$
2: $\mathcal{U} \leftarrow \emptyset$
3: Initialize the prior $\mathcal{P}_0$ with $p_k^0 \leftarrow \frac{1}{n}$, $\forall k = 1, 2, \ldots, n$
4: **repeat**
5:     Compute the sample mean $\bar{\boldsymbol{\mu}}_m$ and the sample covariance $\bar{\boldsymbol{\Sigma}}_m$ from the current belief $\mathcal{P}_m$
6:     Find the largest eigenvalue of $\bar{\boldsymbol{\Sigma}}_m$ and its eigenvector, $\lambda_{\max}$ and $\boldsymbol{v}_{\max}$ respectively
7:     $\tilde{\boldsymbol{z}}_1 \leftarrow \bar{\boldsymbol{\mu}}_m + r \cdot \sqrt{\lambda_{\max}} \boldsymbol{v}_{\max}$
8:     $\tilde{\boldsymbol{z}}_2 \leftarrow \bar{\boldsymbol{\mu}}_m - r \cdot \sqrt{\lambda_{\max}} \boldsymbol{v}_{\max}$
9:     Find two objects $i \neq j$, s.t.

$$i = \underset{i \in [n], i \notin \mathcal{U}}{\arg\min} p_i^m \|\boldsymbol{x}_i - \tilde{\boldsymbol{z}}_1\|_2,$$

$$j = \underset{j \in [n], j \notin \mathcal{U}}{\arg\min} p_j^m \|\boldsymbol{x}_j - \tilde{\boldsymbol{z}}_2\|_2$$

10:     $\mathcal{U} \leftarrow \mathcal{U} \cup \{i, j\}$
11:     Obtain the response $\hat{y}$ from the user
12:     Update belief $\mathcal{P}_{m+1} \leftarrow$ UPDATE$(\mathcal{P}_m, \hat{y})$ using Bayes rule
13:     $m \leftarrow m + 1$
14: **until** $t \in \{i, j\}$

---

Since $\boldsymbol{x}_t$ is not known by the system during the search, we take a Bayesian approach to model the probability of the objects in $[n] = \{1, 2, \ldots, n\}$ to be the target, and at each step $m$ of the search maintain a full belief $\mathcal{P}^m = [p_1^m, p_2^m, \ldots, p_n^m]$ over all $n$ objects. We start with a uniform prior $\mathcal{P}_0 = [\frac{1}{n}, \frac{1}{n}, \ldots, \frac{1}{n}]$.

**Choosing the next query to ask the user.** Similarly to GAUSSSEARCH, at each step we would like to ask a query $(i, j)$ that would maximize the *expected information gain* given the current posterior belief $\mathcal{P}_m$ at step $m$ of the search:

$$(i, j) := \max_{i \neq j} \left( H(\mathcal{P}_m) - \mathbb{E}_{Y|\boldsymbol{x}_i, \boldsymbol{x}_j}[H(\mathcal{P}_m \mid Y)] \right), \tag{3}$$

where $Y \sim P(Y|\boldsymbol{x}_i, \boldsymbol{x}_j)$ is the marginalized belief over the answers to the query $(i, j)$, i.e.

$$P(Y = i \mid \boldsymbol{x}_i, \boldsymbol{x}_j) = \sum_{k=1}^n p_{\boldsymbol{x}_i, \boldsymbol{x}_j, \boldsymbol{x}_k} \, p_k^m.$$

Performing an exhaustive search over all $O(n^2)$ possible pairs $(i, j)$ in order to find the optimal query in terms of (3) would be prohibitively slow, so we propose an alternative heuristic that has good performance in practice.

We first detect the direction along which the variance of the belief is maximized, for that a sample mean and a covariance matrix $(\bar{\boldsymbol{\mu}}_m, \bar{\boldsymbol{\Sigma}}_m)$ are computed from the current belief $\mathcal{P}_m$. Next we build a proto-query as a pair of two points $(\tilde{\boldsymbol{z}}_1, \tilde{\boldsymbol{z}}_2)$ in $\mathbb{R}^d$ that lie in the direction of the maximum variance of $\bar{\boldsymbol{\Sigma}}_m$ on opposite sides of the sample mean $\bar{\boldsymbol{\mu}}_m$. In order to have a desired explore-exploit trade-off of a query, we control the distance from $\tilde{\boldsymbol{z}}_j$ to $\bar{\boldsymbol{\mu}}_m$ by a multiplication parameter $r \in \mathbb{R}_+$. Finally, we find two distinct objects $(i, j)$ from $[n]$ which have the closest representations to $(\tilde{\boldsymbol{z}}_1, \tilde{\boldsymbol{z}}_2)$ in a $\mathcal{P}_m$-weighted Euclidean distance, which favors the near and more probable points. This pair $(i, j)$ becomes the next query to the oracle.

**Posterior UPDATE.** After we obtain the response from the user, $\hat{y} \in \{i, j\}$, the posterior probabilities are updated using Bayes rule $p_k^{m+1} = p_k^m \, P(Y = \hat{y} \mid \boldsymbol{x}_i, \boldsymbol{x}_j)/C$, $k = 1, 2, \ldots, n$, where $C = \sum_{k=1}^n p_k^m \, P(Y = \hat{y} \mid \boldsymbol{x}_i, \boldsymbol{x}_j, \boldsymbol{x}_k)$ is the normalizing costant.

The search finishes when the user indicates one of the query objects as his target, otherwise both query objects are considered to be non-target and further do not appear in the search. We keep track of the objects that we have displayed to the user already using the set of "used" objects $\mathcal{U}$. The complete search algorithm is outlined in Algorithm 3.

The complexity of each step of the Algorithm 3 is $O(nd + d^2)$, since computing the sample covariance is $O(nd)$ and finding the principle eigenvector can be approximated with the power method in $O(d^2)$. Since in practice the number of features $d$ remains constant, the complexity is linear in the number of objects $n$.

**Additional comments on the face search experiment.** In total, we recruited 24 human participants. We presented 10 different target actors to each participant and asked to search for them. We performed an A/B testing by privily using $\gamma$-CKLSEARCH in the backend of the search interface for one half of the searches and GAUSSSEARCH for the other half. The target actors were chosen uniformly at random from a filtered set of 387 actors that had at least 100 associated triplets in $\mathcal{T}$. The A/B testing assignments were designed such that almost all of the chosen targets were paired exactly once with $\gamma$-CKLSEARCH and exactly once with GAUSSSEARCH. Overall the participants did 207 searches with 129 unique targets, 104 searches using GaussSearch and 103 searches using $\gamma$-CKLSEARCH. 19 participants completed all 10 searches, 1 participant completed 7 seaches, 1 participant completed 5 searches, 1 participant completed 3 searches, and 2 participants completed only 1 search. Based on the initial trial runs we ended up with the following choice of hyperparameters: $D = 5$, $\gamma = 3$, $r = 2$ and $\sigma_\varepsilon = 0.1$.

To ensure fair payment, we estimated the duration of our study in trial runs. Participants were paid the equivalent of 20 USD per hour. We do not collect any sensitive data, in particular we do not collect any data that makes a participant personally identifiable. The study design has been reviewed and approved by our IRB.

**Instructions given to participants** The text below is a copy of the instructions given to our participants.

"Want to do a paid search? Here is the way!"

With <name withheld for double-blind review>, you can find that actor or actress interactively! We will show you four faces, and all you have to do is to click on the one who looks most like the person you have in mind. Just repeat this process a few times, until your target appears among the four faces. Click Found to take you to their details.

- Create an account
- Come back here
- Start to make searches!

Are you registered? If yes, start to make searches! Confidentiality:

In accordance with GDPR and European laws on privacy, our website uses cookies. However, only necessary cookies are used (to identify you and let you perform your search). If you chose to refuse the use of cookies, you won't be able to use our website. We will not share personally identifiable information with anyone. However, we may use anonymized and aggregated information collected from this experiment for research purposes, and potentially release such information publicly in the spirit of Open Science and reproducibility.

## C THEOREM 2.1

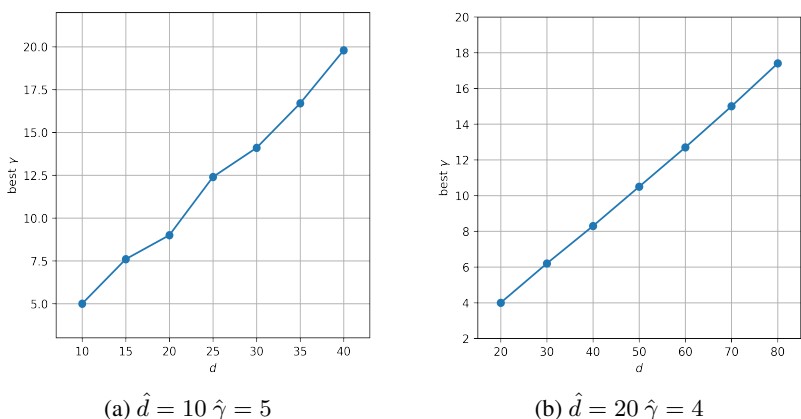

(a) $\hat{d} = 10 \ \hat{\gamma} = 5$          (b) $\hat{d} = 20 \ \hat{\gamma} = 4$

Figure 7: Linear relationship between $\gamma$ and $d$ for finite values of $d$.

**Experiments on the relationship between $\gamma$ and $d$.** In our experiment first we fix the reference values of $\hat{d}$ and $\hat{\gamma}$ for which compute the average probability of the correct answer $p_Q(\hat{\gamma}, \hat{d})$. Then we iterate over the values of $d > \hat{d}$ and for each we

find the corresponding $\gamma$ that minimizes $|p_Q(\hat{\gamma}, \hat{d}) - p_Q(\gamma, d)|$ via a gridsearch. The best values of $\gamma$ are reported in Fig. 7. In all trials we kept $N = 1000$ and $|\mathcal{T}| = 10'000$. We observe a linear relationship between $\gamma$ and $d$ even for finite values of $d$, which matches the limit result of Theorem 2.1.

*Proof of Theorem 2.1.* Consider two points $x_a, x_b \in \mathbb{R}^d$ sampled uniformly from a unit ball $\mathcal{B}$ that form a query to the oracle $Q = (x_a, x_b)$. After asking $Q$ we observe the answer $Y \in \{x_a, x_b\}$ under the $\gamma$-CKL model for some fixed $\gamma \geq 2$. Then the probability that the answer $Y$ is correct, $p_Q$, is

$$
\begin{aligned}
p_Q &= \int_{r_1=0}^{1} \int_{r_2=r_1}^{1} \frac{r_2^\gamma}{r_1^\gamma + r_2^\gamma} S_d(r_1) S_d(r_2) \frac{1}{V_d} \frac{1}{V_d} dr_1 dr_2 \\
&\quad + \int_{r_1=0}^{1} \int_{r_2=0}^{r_1} \frac{r_1^\gamma}{r_1^\gamma + r_2^\gamma} S_d(r_1) S_d(r_2) \frac{1}{V_d} \frac{1}{V_d} dr_1 dr_2 \\
&= \int_{r_1=0}^{1} \int_{r_2=r_1}^{1} \frac{r_2^\gamma}{r_1^\gamma + r_2^\gamma} r_1^{d-1} r_2^{d-1} d^2 dr_1 dr_2 \qquad (4) \\
&\quad + \int_{r_1=0}^{1} \int_{r_2=0}^{r_1} \frac{r_1^\gamma}{r_1^\gamma + r_2^\gamma} r_1^{d-1} r_2^{d-1} d^2 dr_1 dr_2, \qquad (5)
\end{aligned}
$$

where

$$
S_d(r) = \frac{2\pi^{\frac{d}{2}}}{\Gamma(\frac{d}{2})} r^{d-1}, \quad V_d = \frac{\pi^{\frac{d}{2}}}{\Gamma(\frac{d}{2} + 1)}
$$

are the respective surface and volume of the unit ball $\mathcal{B}$.

Consider (4),

$$
\int_{r_1=0}^{1} \int_{r_2=r_1}^{1} \frac{r_2^\gamma}{r_1^\gamma + r_2^\gamma} r_1^{d-1} r_2^{d-1} d^2 dr_1 dr_2.
$$

If we increase $d$, the distance from the center of the ball to a random inside point will be close to 1. We use Taylor approximation of the probability model at $(1, 1) \in \mathbb{R}^2$:

$$
\begin{aligned}
\frac{r_2^\gamma}{r_1^\gamma + r_2^\gamma} &= \frac{1}{2} - (r_1 - 1)\frac{\gamma}{4} + (r_2 - 1)\frac{\gamma}{4} + R(r_1, r_2) \\
&= P(r_1, r_2) + R(r_1, r_2).
\end{aligned}
$$

Let's fix some $0 < \varepsilon < 1$. Then

$$
\begin{aligned}
(4) &= \int_{r_1=0}^{1} \int_{r_2=r_1}^{1} \frac{r_2^\gamma}{r_1^\gamma + r_2^\gamma} r_1^{d-1} r_2^{d-1} d^2 dr_1 dr_2 \\
&= \int_{r_1=\varepsilon}^{1} \int_{r_2=r_1}^{1} \frac{r_2^\gamma}{r_1^\gamma + r_2^\gamma} r_1^{d-1} r_2^{d-1} d^2 dr_1 dr_2 \\
&\quad + \int_{r_1=0}^{\varepsilon} \int_{r_2=r_1}^{1} \frac{r_2^\gamma}{r_1^\gamma + r_2^\gamma} r_1^{d-1} r_2^{d-1} d^2 dr_1 dr_2 \\
&= \int_{r_1=\varepsilon}^{1} \int_{r_2=r_1}^{1} P(r_1, r_2) r_1^{d-1} r_2^{d-1} d^2 dr_1 dr_2 \\
&\quad + \int_{r_1=\varepsilon}^{1} \int_{r_2=r_1}^{1} R(r_1, r_2) r_1^{d-1} r_2^{d-1} d^2 dr_1 dr_2 \\
&\quad + \int_{r_1=0}^{\varepsilon} \int_{r_2=r_1}^{1} \frac{r_2^\gamma}{r_1^\gamma + r_2^\gamma} r_1^{d-1} r_2^{d-1} d^2 dr_1 dr_2.
\end{aligned}
$$

First note that the last summand is $o(1)$ when $d \to \infty$:

$$\int_{r_1=0}^{\varepsilon} \int_{r_2=r1}^{1} \frac{r_2^{\gamma}}{r_1^{\gamma} + r_2^{\gamma}} r_1^{d-1} r_2^{d-1} d^2 dr_1 dr_2$$

$$\leq \int_{r_1=0}^{\varepsilon} \int_{r_2=r1}^{1} r_1^{d-1} r_2^{d-1} d^2 dr_1 dr_2$$

$$\leq \frac{1}{2}\varepsilon^d (2 - \varepsilon^d) = o(1).$$

Now the integral with the $P(r_1, r_2)$ term can be computed as follows:

$$\int_{r_1=\varepsilon}^{1} \int_{r_2=r1}^{1} P(r_1, r_2) r_1^{d-1} r_2^{d-1} d^2 dr_1 dr_2 =$$

$$= \int_{r_1=0}^{1} \int_{r_2=r1}^{1} \frac{1}{2} r_1^{d-1} r_2^{d-1} d^2 dr_1 dr_2 + \frac{1}{4}\varepsilon^d - \frac{1}{2}\varepsilon^{2d}$$

$$+ \int_{r_1=0}^{1} \int_{r_2=r1}^{1} (r_1 - 1)\frac{\gamma}{4} r_1^{d-1} r_2^{d-1} d^2 dr_1 dr_2$$

$$+ \frac{\gamma d\varepsilon^d}{4} \left( \frac{\varepsilon^2 - 2}{2d} + \varepsilon \left( \frac{1}{d+1} - \frac{\varepsilon^d}{2d+1} \right) \right)$$

$$+ \int_{r_1=0}^{1} \int_{r_2=r1}^{1} (r_2 - 1)\frac{\gamma}{4} r_1^{d-1} r_2^{d-1} d^2 dr_1 dr_2$$

$$+ \frac{\gamma \varepsilon^d (d(2d(\varepsilon - 1) - 3) - 1)\varepsilon^d + 2(2d+1)}{8(d+1)(2d+1)}$$

$$= \frac{1}{4} + \frac{\gamma}{4} \frac{d^2(3d+1)}{2d^2(d+1)(2d+1)} - \frac{\gamma}{4} \frac{d^2}{4d^3 + 2d^2} + o(1)$$

$$= \frac{1}{4} + \frac{\gamma}{4} \frac{d}{(d+1)(2d+1)} + o(1).$$

Finally, consider the remaining integral,

$$\int_{r_1=\varepsilon}^{1} \int_{r_2=r1}^{1} R(r_1, r_2) r_1^{d-1} r_2^{d-1} d^2 dr_1 dr_2.$$

Using Taylor's theorem for multivariate functions, we can get an upper bound for its absolute value:

$$\left| \int_{r_1=\varepsilon}^{1} \int_{r_2=r1}^{1} R(r_1, r_2) r_1^{d-1} r_2^{d-1} d^2 dr_1 dr_2 \right|$$

$$\leq \frac{M(\gamma)}{2} \int_{\mathcal{X}} \left( (r_1 - 1)^2 + (r2 - 1)^2 \right) r_1^{d-1} r_2^{d-1} d^2 dr_1 dr_2$$

$$+ \frac{M(\gamma)}{2} \int_{\mathcal{X}} 2(r_1 - 1)(r_2 - 1) r_1^{d-1} r_2^{d-1} d^2 dr_1 dr_2$$

where

$$M(\gamma) = \max_{\alpha = |2|, (r_1, r_2) \in \mathcal{X}} \left| D^{\alpha} \left[ \frac{r_2^{\gamma}}{r_1^{\gamma} + r_2^{\gamma}} \right] \right|,$$

$$\mathcal{X} = \{ (r_1, r_2) \mid r_1 \in [\varepsilon, 1], \ r_2 \in [r_1, 1] \},$$

and

$$\left| D^{(2,0)} \left[ \frac{r_2^\gamma}{r_1^\gamma + r_2^\gamma} \right] \right| = \left| \frac{\gamma r_2^{\gamma-2} r_1^\gamma ((\gamma-1)r_1^\gamma - (\gamma+1)r_2^\gamma)}{(r_1^\gamma + r_2^\gamma)^3} \right|,$$

$$\left| D^{(1,1)} \left[ \frac{r_2^\gamma}{r_1^\gamma + r_2^\gamma} \right] \right| = \frac{\gamma^2 r_2^{\gamma-1} r_1^{\gamma-1} (r_2^\gamma - r_1^\gamma)}{(r_1^\gamma + r_2^\gamma)^3},$$

$$\left| D^{(0,2)} \left[ \frac{r_2^\gamma}{r_1^\gamma + r_2^\gamma} \right] \right| = \left| \frac{\gamma r_2^\gamma r_1^{\gamma-2} ((\gamma+1)r_1^\gamma) - (\gamma-1)r_2^\gamma}{(r_1^\gamma + r_2^\gamma)^3} \right|.$$

For a big enough $d$, if $\gamma$ grows with $d$, the maximum of $M(\gamma)$ is achieved when $r_1 = r_2$ with $M(\gamma) \le \frac{\gamma}{4}\varepsilon^{-2}$. We will show this for $\left| D^{(2,0)} \left[ \frac{r_2^\gamma}{r_1^\gamma + r_2^\gamma} \right] \right|$, the other two cases can be proved similarly. Indeed,

$$\begin{aligned}
\left| D^{(2,0)} \left[ \frac{r_2^\gamma}{r_1^\gamma + r_2^\gamma} \right] \right| &= \left| \frac{\gamma r_2^{\gamma-2} r_1^\gamma ((\gamma-1)r_1^\gamma - (\gamma+1)r_2^\gamma)}{(r_1^\gamma + r_2^\gamma)^3} \right| \\
&= \gamma \frac{r_2^{\gamma-2} r_1^\gamma ((\gamma+1)r_2^\gamma - (\gamma-1)r_1^\gamma)}{(r_1^\gamma + r_2^\gamma)^3} \\
&= \gamma \frac{\left(\frac{r_2}{r_1}\right)^\gamma \left((\gamma+1)\left(\frac{r_2}{r_1}\right)^\gamma - (\gamma-1)\right)}{r_2^2 (1 + \left(\frac{r_2}{r_1}\right)^\gamma)^3},
\end{aligned}$$

which is equal to $\frac{\gamma}{4}\varepsilon^{-2}$ when $r_1 = r_2$ and goes to 0 with $d \to \infty$ when $r_1 < r_2$.

Finally

$$\int_{\mathcal{X}} \left( (r_1 - 1)^2 + 2(r_1 - 1)(r_2 - 1) + (r2 - 1)^2 \right) r_1^{d-1} r_2^{d-1} d^2 dr_1 dr_2$$
$$= \varepsilon^{2d} P_1 + \varepsilon^d P_2 + \frac{3d+4}{(d+1)^2(d+2)},$$

where

$$P_1 = -\frac{d \left( d^2 + (d+1)^2 \varepsilon^2 - 2(d+2)^2 \varepsilon + 6d + 13 \right) + 8}{(d+1)^2(d+2)}$$

and

$$P_2 = \frac{(2(d+2)d+1)d\varepsilon^2 - 4d(d+2)(d+1)\varepsilon + 2(d+2)(d+1)^2}{(d+1)^2(d+2)}$$

are two polynomial fractions.

Putting everything together we can upper bound the remainder by

$$\left| \int_{r_1=\varepsilon}^1 \int_{r_2=r1}^1 R(r_1,r_2) r_1^{d-1} r_2^{d-1} d^2 dr_1 dr_2 \right|$$
$$\le \frac{\gamma \varepsilon^{-2}}{8} \left( \varepsilon^{2d} P_1 + \varepsilon^d P_2 + \frac{3d+4}{(d+1)^2(d+2)} \right).$$

Also, due to symmetry, $(4) = (5)$, and thus

$$p_Q = \frac{1}{2} + \frac{\gamma}{2} \frac{d}{(d+1)(2d+1)} + \hat{R} + o(1)$$

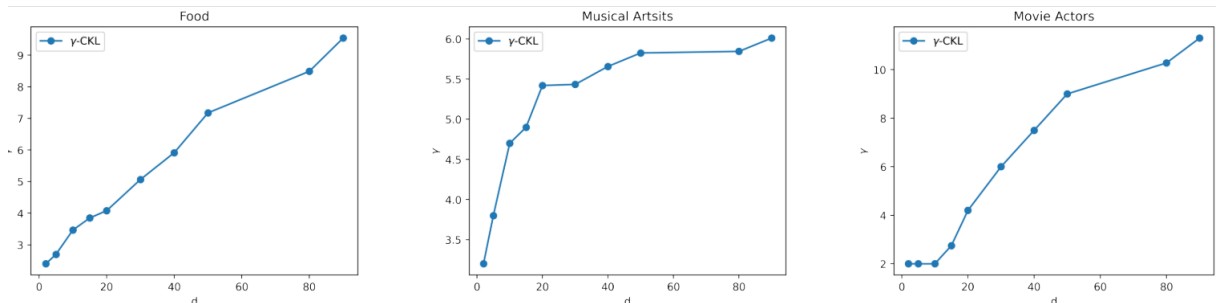

Figure 8: Running average of the best performing values of $\gamma$ in $\gamma$-CKL embedding as we increase the embedding dimensionality $d$.

where

$$|\hat{R}| \leq \frac{\gamma \varepsilon^{-2}}{4}\left(\varepsilon^{2d}P_1 + \varepsilon^d P_2 + \frac{3d+4}{(d+1)^2(d+2)}\right).$$

We see that if

$$\frac{\gamma}{d} = c_1 + o(1)$$

and $d \to \infty$, then

$$p_Q = c_2 + o(1),$$

where $c_1 > 0$, $c_2 > 0$ are constants. $\qquad \square$

## D   $\gamma$-$d$ RELATION IN THE EMBEDDING EXPERIMENTS

For $\gamma$-CKL as $d$ increases, the best performing values of $\gamma$ tend to also increase, which is aligned with the findings of Theorem 2.1, see Fig. 8. For each dataset and each value of $d$ we report the running average of the mean of the top 10 best performing values of $\gamma$ for that $d$. We see that the running average value of $\gamma$ is uniformly lower for for the Musical Artists dataset than for the other two datasets. We suspect this is because the dataset itself contains relatively small average number of triplets per object. That is why the $\gamma$-CKL embedding does not profit from increasing the values of $\gamma$, which could lead the model to be more confident when predicting outcome probabilities.

## E   PROOF OF PROPOSITION 2.2

*Proof.* First we will show that for a query $Q_i = (\boldsymbol{x}_i^a, \boldsymbol{x}_i^b)$ the set of points $\mathcal{S}_i \subset \Omega$ for which the expected log-likelihood of the answer $Y$ is maximized forms a $d$-dimensional sphere. For ease of reading, we drop the index $i$ and simply write $Q = (\boldsymbol{x}_a, \boldsymbol{x}_b)$.

The oracle will answer $Y = \boldsymbol{x}_a$ with probability $p(\boldsymbol{x}_a, \boldsymbol{x}_b; \boldsymbol{x}_t) = \frac{\|\boldsymbol{x}_b - \boldsymbol{x}_t\|^\gamma}{\|\boldsymbol{x}_a - \boldsymbol{x}_t\|^\gamma + \|\boldsymbol{x}_b - \boldsymbol{x}_t\|^\gamma}$. Then all points $\boldsymbol{x} \in \Omega$ s.t. $p_{\boldsymbol{x}_a, \boldsymbol{x}_b, \boldsymbol{x}} = p_{\boldsymbol{x}_a, \boldsymbol{x}_b, \boldsymbol{x}_t}$ will have the largest expected log-likelihood values. Now denoting

$$c := \frac{\|\boldsymbol{x}_a - \boldsymbol{x}_t\|^2}{\|\boldsymbol{x}_b - \boldsymbol{x}_t\|^2} = \left(\frac{1}{p} - 1\right)^{\frac{2}{\gamma}},$$

and observing

$$p = \frac{\|\boldsymbol{x}_b - \boldsymbol{x}_t\|^\gamma}{\|\boldsymbol{x}_a - \boldsymbol{x}_t\|^\gamma + \|\boldsymbol{x}_b - vx_t\|^\gamma} = \frac{1}{\frac{\|\boldsymbol{x}_a - \boldsymbol{x}_t\|^\gamma}{\|\boldsymbol{x}_b - \boldsymbol{x}_t\|^\gamma} + 1},$$

we can define the set $\mathcal{S}_Q$ by

$$\sum_{j=1}^{d}(\boldsymbol{x}_j - (\boldsymbol{x}_a)_j)^2 - c\sum_{i=j}^{D}(\boldsymbol{x}_j - (\boldsymbol{x}_b)_j)^2 = 0,$$

which is equivalent to

$$\sum_{i=j}^{d}(\boldsymbol{x}_j - \boldsymbol{z}_j)^2 = r.$$

for

$$z_j = \frac{c(\boldsymbol{x}_b)_j - (\boldsymbol{x}_a)_j}{1-c},$$

$$r = \sum_{j=1}^{d}\frac{(c(\boldsymbol{x}_b)_j - (\boldsymbol{x}_a)_j)^2}{(1-c)^2} - \frac{(\boldsymbol{x}_a)_j^2 - c(\boldsymbol{x}_b)_j^2}{(1-c)}.$$

Hence, for a fixed query, the points that have the same likelihood as $\boldsymbol{x}_t$ (and which will have the maximal expected log-likelihood) form a sphere in $\mathbb{R}^d$. For two distinct query points $Q_1$ and $Q_2$, the set of points with maximal expected log-likelihood for $Q_1$ and $Q_2$ will lie in the intersection of $\mathcal{S}_1$ and $\mathcal{S}_2$, i.e., at least in a $(d-1)$-dimensional sphere $\mathcal{S}_1 \cap \mathcal{S}_2$. Consider the third query point $Q_3$. The intersection $\mathcal{S}_1 \cap \mathcal{S}_2 \cap \mathcal{S}_3$ is at least a $(d-2)$-dimensional sphere if $\boldsymbol{z}_3$ does not lie on the line intersecting $\boldsymbol{z}_1$ and $\boldsymbol{z}_2$, otherwise the points in $\mathcal{S}_1 \cap \mathcal{S}_2$ are equidistant from $\boldsymbol{z}_3$, and since $\boldsymbol{x}_t \in \mathcal{S}_1 \cap \mathcal{S}_2$, $\mathcal{S}_1 \cap \mathcal{S}_2 = \mathcal{S}_1 \cap \mathcal{S}_2 \cap \mathcal{S}_3$, and no additional dimensionality reduction of the spheres intersection is achieved (see Fig 1 for illustration). Similarly, for $d+1$ queries $Q_1, Q_2, \ldots, Q_{d+1}$, the sufficient condition for

$$\mathcal{S}_1 \cap \mathcal{S}_2 \cap \cdots \cap \mathcal{S}_{d+1} = \boldsymbol{x}_t$$

is

$$\text{rank}(\tilde{\boldsymbol{z}}_1 - \tilde{\boldsymbol{z}}_{d+1}, \tilde{\boldsymbol{z}}_2 - \tilde{\boldsymbol{z}}_{d+1}, \ldots, \tilde{\boldsymbol{z}}_d - \tilde{\boldsymbol{z}}_{d+1}) = d. \tag{6}$$

The intersection of the $d+1$ corresponding spheres will result in exactly one point, $\boldsymbol{x}_t$. Thus the expected log-likelihood after $d+1$ such queries will be maximized only at $\boldsymbol{x}_t$. Now by chosing a uniform prior over $\Omega$, in expectation over the outcomes of any set of queries $\tilde{\mathcal{Q}}$ that satisfies (6) the posterior will be maximized only at $\boldsymbol{x}_t$ and then the claim follows immediately. $\qquad\square$

# F   FULL PROOFS FOR EXPONENTIAL CONVERGENCE

*Proof for Lemma 3.1.* Let $\mathcal{D}$ be the decision made by the algorithm. This is a random variable which can take values in $(B, P_1, P_2, \ldots, P_{5^d})$, for backtracking or proceeding to one of the children of $X$. When arriving at a region $X$, the algorithm discards information about previous query outcomes. Then it asks a series of queries, prescribed by our Inner Loop algorithm. Once the query outcomes have been observed, the decision is deterministic. Therefore, to describe the distribution of $\mathcal{D}$ it is sufficient to describe the distribution of queries and query outcomes. The queries that we ask depend only on the current region. The outcomes are conditionally independent, given a target location. Therefore, the distribution of $\mathcal{D}$ only depends on the current region and the latent $\mathbf{x}_t$. This means that the decision to proceed or backtrack is Markovian. The sequence of regions $X_s$ is a random walk. $\qquad\square$

*Proof for Lemma 3.5.* We show an equivalent statement: There exists a coupling $\tilde{X}_s$ and $\tilde{Z}_s$, such that $\forall s \geq 0 :$ $\mathbb{P}[z(\mathbf{x}_t, \tilde{X}_s) \leq \tilde{Z}_s] = 1$ and the distributions are identical, $F_{X_s} = F_{\tilde{X}_s}, F_{Z_s} = F_{\tilde{Z}_s}$ This is done via induction.

*Induction start:*
For $s = 0$ we are looking at a constant, which is the same in both cases: $Z_0 = z(\mathbf{x}_t, X_0)$. Immediately $\mathbb{P}[z(\mathbf{x}_t, X_0) = \tilde{Z}_0] = 1$

*Induction step:*
We are given a random variable $\tilde{\mathcal{X}}_s$ which has the same distribution as $X_s$ and we know that $\mathbb{P}[z(\mathbf{x}_t, \tilde{X}_s) \leq \tilde{Z}_s] = 1$. We will now construct two random variables $\tilde{\mathcal{X}}_{s+1}$ and $\tilde{Z}_{s+1}$ for which it holds that $\mathbb{P}[z(\mathbf{x}_t, \tilde{X}_{s+1}) \leq \tilde{Z}_{s+1}] = 1$.

Let $u \sim U[0,1]$ be a sample from the uniform distribution on $(0,1)$. We use this to couple the two random walks. Depending on $u$ and the current state of the random walk is $\tilde{X}_s$, the following transition is taken:

- $\tilde{X}_s$ is green. This means $z(\mathbf{x}_t, \tilde{X}_s) = 0$
    - if $u \le p_d(\tilde{X}_s, \mathbf{x}_t)$, then proceed to a green child. This means $z(\mathbf{x}_t, \tilde{X}_{s+1}) = 0$
    - if $p_d(\tilde{X}_s, \mathbf{x}_t) < u \le p_d(\tilde{X}_s, \mathbf{x}_t) + q_u(\tilde{X}_s, \mathbf{x}_t)$, then backtrack to the parent region. This means $z(\mathbf{x}_t, \tilde{X}_{s+1}) = 0$
    - else $p_d(\tilde{X}_s, \mathbf{x}_t) + q_u(\tilde{X}_s, \mathbf{x}_t) < u \le 1$, stray to a red child region. Now we have $z(\mathbf{x}_t, \tilde{X}_{s+1}) = 1$
- $\tilde{X}_s$ is red. This means $z(\mathbf{x}_t, \tilde{X}_s) > 0$
    - if $u \le p_u(\tilde{X}_s, \mathbf{x}_t)$, then backtrack to the parent region. This means $z(\mathbf{x}_t, \tilde{X}_{s+1}) - z(\mathbf{x}_t, \tilde{X}_s) = -1$
    - if $p_u(\tilde{X}_s, \mathbf{x}_t) \le u < p_r \tilde{X}_s, \mathbf{x}_t) + p_u(\tilde{X}_s, \mathbf{x}_t)$, then recover by proceeding to a green child region. This means that $z(\mathbf{x}_t, \tilde{X}_{s+1}) = 0$. Recovering is only possible if one of the child regions contains the target. We know that the parent of a region is a superset of all the child regions $u(X) \supset \bigcup_{X_c \in D(X)} X_c$. Therefore, whenever a recovery transition is possible, backtracking must likewise lead to a green region. This shows that recovery is only possible when $z(\mathbf{x}_t, \tilde{X}_s) = 1$. Therefore we have shown that $z(\mathbf{x}_t, \tilde{X}_{s+1}) - z(\mathbf{x}_t, \tilde{X}_s) = -1$
    - else $p_r \tilde{X}_s, \mathbf{x}_t) + p_u(\tilde{X}_s, \mathbf{x}_t) \le u \le 1$, then proceed to a red child. This means $z(\mathbf{x}_t, \tilde{X}_{s+1}) - z(\mathbf{x}_t, \tilde{X}_s) \in \{0, 1\}$

We now construct a coupled variable $\tilde{D}$ such that $\tilde{Z}_{s+1} = \tilde{Z}_s + \tilde{D}$. Since $\tilde{Z}$ is a random walk on natural numbers, with a self-loop at 0, we need to distinguish between two scenarios:

- $\tilde{Z}_s > 0$
    - if $u \le \frac{1+b}{2}$, then $\tilde{D} = -1$
    - else, $\tilde{D} = 1$
- $\tilde{Z}_s = 0$
    - if $u \le \frac{1+b}{2}$, then $\tilde{D} = 0$
    - else, $\tilde{D} = 1$

We will now show that the construction of $\tilde{D}$ ensures that $\mathbb{P}[z(\mathbf{x}_t, \tilde{X}_{s+1}) \le \tilde{D} + \tilde{Z}_s] = 1$

**Case 1**, $\tilde{Z}_s = 0$:

The induction assumptions imply $z(\mathbf{x}_t, \tilde{X}_s) = 0$, which in turn implies that $\tilde{\mathcal{X}}_s$ is a green region. In this case we know that $\tilde{D} \in \{0, 1\}$ and $z(\mathbf{x}_t, \tilde{X}_{s+1}) \in \{0, 1\}$.

It holds that $z(\mathbf{x}_t, \tilde{X}_{s+1}) = 1$ iff $p_d(\tilde{X}_s, \mathbf{x}_t) + q_u(\tilde{X}_s, \mathbf{x}_t) = 1 - q_s(\tilde{\mathcal{X}}_s, \mathbf{x}_t) < u$. It holds that $\tilde{D} = 1$ iff $\frac{1+b}{2} < u$. From assumption 3.2 we know that for all possible regions $X$ and targets $\mathbf{x}_t$, $1 - q_s(X, \mathbf{x}_t) > \frac{1+b}{2}$. Therefore $\tilde{D} = 0 \implies z(\mathbf{x}_t, \tilde{X}_{s+1}) = 0$. Therefore $\mathbb{P}[z(\mathbf{x}_t, \tilde{X}_{s+1}) \le \tilde{D} + \tilde{Z}_s \mid \tilde{Z}_s = 0] = 1$

**Case 2**, $\tilde{Z}_s > 0, z(\mathbf{x}_t, \tilde{X}_s) = 0$:

Again, $\tilde{X}_s$ is a green region. So we know that $z(\mathbf{x}_t, \tilde{X}_{s+1}) \in \{0, 1\}$, and $z(\mathbf{x}_t, \tilde{X}_{s+1}) = 1$ iff $p_d(\tilde{X}_s, \mathbf{x}_t) + q_u(\tilde{X}_s, \mathbf{x}_t) = 1 - q_s(\tilde{\mathcal{X}}_s, \mathbf{x}_t) < u$.

Since $\tilde{Z}_s > 0$ and $z(\mathbf{x}_t, \tilde{X}_s) = 0$ we know $\tilde{D} \ge 0 \implies z(\mathbf{x}_t, \tilde{X}_{s+1}) \le \tilde{Z}_s + \tilde{D}$. We only need to analyse the case of $\tilde{D} = -1$. We know that $\tilde{D} = -1 \implies u < \frac{1+b}{2}$. Using assumption 3.2, $z(\mathbf{x}_t, \tilde{X}_{s+1}) = 1 \implies u > 1 - p_s(\tilde{X}_s, \mathbf{x}_t) > 1 - \frac{1-b}{2} = \frac{1+b}{2}$. This is a contradiction. We now know that $\tilde{D} = -1 \implies z(\mathbf{x}_t, \tilde{X}_{s+1}) = 0$. Therefore $\mathbb{P}[z(\mathbf{x}_t, \tilde{X}_{s+1}) \le \tilde{D} + \tilde{Z}_s \mid \tilde{Z}_s > 0, z(\mathbf{x}_t, \tilde{X}_s) = 0] = 1$

**Case 3**, $\tilde{Z}_s > 0, z(\mathbf{x}_t, \tilde{X}_s) > 0$:

$\tilde{D} = -1$ implies $u < \frac{1+b}{2}$. From Assumption 3.2 we know that $\forall X, \mathbf{x}_t : \frac{1+b}{2} \le p_u(X_s, \mathbf{x}_t) + p_r X_s, \mathbf{x}_t)$. Therefore the event $\tilde{D} = -1$ implies $z(\mathbf{x}_t, \tilde{X}_{s+1}) - z(\mathbf{x}_t, \tilde{X}_s) = -1$. Therefore $\mathbb{P}[z(\mathbf{x}_t, \tilde{X}_{s+1}) \le \tilde{D} + \tilde{Z}_s \mid \tilde{Z}_s > 0, z(\mathbf{x}_t, \tilde{X}_s) > 0] = 1$

We have shown that $\mathbb{P}[z(\mathbf{x}_t, \tilde{X}_{s+1}) \le \tilde{D} + \tilde{Z}_s] = 1$ ◻

*Proof for Lemma 3.6.* Under the assumption 3.2 we have shown $z(\mathbf{x}_t, X_s) \preceq_{st.} Z_s$. The definition of stochastic ordering $\mathbb{P}[z(\mathbf{x}_t, X_s) \ge k] \le \mathbb{P}[Z_s \ge k]$ is equivalent to $\mathbb{P}[z(\mathbf{x}_t, X_s) \le k] \ge \mathbb{P}[Z_s \le k]$.

We will now show the claim of the lemma for $Z_s$, the same statement for $z(\mathbf{x}_t, X_s)$ follows immediately. Our proof is an induction for $\mathbf{P}[Z_s > k] \le (\frac{1-b}{1+b})^k$. The property holds trivially for $s = 0, k \ge 0$ and $k = 0, s \ge 0$. Assume that the

property holds for a given $s$ and for all $k$. For any $k \geq 1$, we have

$$\mathbf{P}[Z_{s+1} > k] = \frac{1+b}{2}\mathbf{P}[Z_s > k+1] + \frac{1-b}{2}\mathbf{P}[Z_s > k-1]$$
$$\leq \frac{1+b}{2}(\frac{1-b}{1+b})^{k+1} + \frac{1-b}{2}(\frac{1-b}{1+b})^{k-1} = (\frac{1-b}{1+b})^k$$

$\square$

*Proof for Lemma 3.7.* Let $\tau_{Z=N} = \inf\{s > 0 \mid Z_s, Z_0 = N\}$ be the stopping time of $Z_s$ reaching 0, starting from $N$. We have shown that the random walk $Z$ can be used as a stochastic upper bound. Therefore we know $\mathbb{E}[\tau_X] \leq \mathbb{E}[\tau_{Z=1}]$.

We will now calculate the stopping time of $Z$.

We ascertain that this random walk is ergodic Levin and Peres [2017]. Since $b > 0$ the random walk is positive recurrent. The self-loop at $Z = 0$ makes it aperiodic. It is irreducible.

Therefore it has a unique stationary distribution $\pi$. We now calculate $\pi$. The conditions on the distribution are:

$$(1), \pi_0 = \frac{1+b}{2}\pi_0 + \frac{1+b}{2}\pi_1$$
$$(2), \pi_1 = \frac{1-b}{2}\pi_0 + \frac{1+b}{2}\pi_2$$
$$(3), \pi_n = \frac{1-b}{2}\pi_{n-1} + \frac{1+b}{2}\pi_{n+1}, n > 1$$
$$\sum_{i=0}^{\infty} \pi_i = 1$$

We show $\pi_n = (\frac{1-b}{1+b})^n\pi_0, n > 0$ by induction:

$$(1) \iff \pi_1 = \frac{1-b}{1+b}\pi_0$$
$$(1)\&(2) \iff \pi_1 = \frac{1-b}{2}\pi_0 + \frac{1+b}{2}\pi_2$$
$$\iff \frac{2}{1+b}\pi_1 - \frac{1-b}{1+b}\pi_0 = \pi_2 \iff \pi_2 = (\frac{1-b}{1+b})^2\pi_0$$
$$(3) \iff \pi_{n-1} = \frac{1-b}{2}\pi_{n-2} + \frac{1+b}{2}\pi_n$$
$$\iff (\frac{1-b}{1+b})^{n-1}\pi_0 - \frac{1-b}{2}(\frac{1-b}{1+b})^{n-2}\pi_0 = \frac{1+b}{2}\pi_n$$
$$\iff (\frac{1-b}{1+b} - \frac{1-b}{2})(\frac{1-b}{1+b})^{n-2}\pi_0 = \frac{1+b}{2}\pi_n$$
$$\iff (\frac{1-b}{1+b}\frac{2}{1+b} - \frac{1-b}{2}\frac{2}{1+b})(\frac{1-b}{1+b})^{n-2}\pi_0 = \pi_n$$
$$\iff (\frac{2-2b-1+b^2}{(1+b)^2})(\frac{1-b}{1+b})^{n-2}\pi_0 = \pi_n$$
$$\iff (\frac{1-b}{1+b})^n\pi_0 = \pi_n$$

From the infinite sum we get $\sum_{i=0}^{\infty} \pi_i = \pi_0 \frac{1}{1-\frac{1-b}{1+b}} = \pi_0\frac{b+1}{2b}$. Therefore: $\pi_0 = \frac{2b}{b+1}$.

We can use the unique stationary distribution of $Z$ to compute expected return times. As defined above, let $\tau_{Z=N} = \inf\{s > 0 \mid Z_s = 0, Z_0 = N\}$. We know that for a unique stationary distribution, the expected inter-arrival time for state 0 is $\mathbb{E}[\tau_{Z=0}] = \frac{1}{\pi_0} = \frac{b+1}{2b}$. We are interested in $\mathbb{E}[\tau_{Z=1}]$. Starting from $Z = 0$ the walk must either follow the self loop or go to

$Z = 1$. This leads to the following equation:

$$\frac{1+b}{2} + \frac{1-b}{2}(\mathbb{E}[\tau_{Z=1}] + 1) = \frac{1}{\pi_0} = \frac{b+1}{2b}$$

$$\implies \frac{1-b}{2}\mathbb{E}[\tau_{Z=1}] = \frac{b+1}{2b} - \frac{1-b}{2}$$

$$+ \frac{1+b}{2} = \frac{b+1}{2b}$$

$$\implies \mathbb{E}[\tau_{Z=1}] = \left(\frac{1}{b}\right)$$

$\square$

*Proof for Theorem 3.8.* The first part of the claim follows immediately from Lemma 3.6. At any time $s$, the probability of needing more than $k$ backtracks until we reach a green region from $X_s$ is less than $\left(\frac{1-b}{1+b}\right)^k$. We solve $\delta = \left(\frac{1-b}{1+b}\right)^k$ for $k$. Then we know that with probability of at least $1 - \delta$, the target must be in the $k$-th ancestor of $X_s$. This is the region that we propose as the result of our search process.

We now need to show that the expected depth of this region increases at a constant rate. Since $k$ is a constant that only depends on the desired rate of error $\delta$ and does not change over time, it suffices to show that the expected depth of $X_s$ increases at a constant rate.

We make use of the Markovian property of $X_s$. Without loss of generality, we assume that we are currently at time $s = 0$. Additionally we assume that the current region $X_0$ is green. When the execution of the algorithm begins, this is true since $\mathbf{x}_t \in \Omega$.

We now define a stopping time $s' = \inf\{s > 0 \mid \mathbf{x}_t \in X_s, X_0\}$, as the next time at which our algorithm visits a green region. We will show that this stopping time is finite and that this next green node is, in expectation, at a higher depth. The analysis then becomes recursive. Specifically, we will show:

- There is a constant $C_d > 0$, such that $\mathbb{E}[D(X_{s'}) - D(X_0)] > C_d$
- There is a constant $C_s < \infty$, such that $\mathbb{E}[s'] < C_s$

Starting from the green region $X_0$, the following transitions are possible:

- With probability $p_d(X_0, \mathbf{x}_t)$, the search proceeds to a green child. In this case we stop immediately, $s' = 1$ and $D(X_1) - D(X_0) = 1$.
- With probability $q_u(X_0, \mathbf{x}_t)$, the search backtracks. Since the parent of a green region must be green as well, we also stop immediately, $s' = 1$. Since backtracking looses two levels of depth, we have $D(X_1) - D(X_0) = -2$.
- With probability $q_s(X_0, \mathbf{x}_t)$, the search strays.

The last case requires further analysis. Following Lemma 3.7, we know that the expected stopping time after straying is upper bounded by $\mathbb{E}[\tau_{Z=1}] = \frac{1}{b}$. Every backtracking decision must always undo at least one proceed decision. This means that, in the worst case scenario, exactly half the steps until $s'$ are proceed and half are backtrack decisions. A pair of proceed and backtrack decisions first gains one level of depth and then looses two. Therefore, conditioned on the assumption that we have left $X_0$ by straying, the expected new depth is bounded by $\mathbb{E}[D(X_{s'}) - D(X_0)|\text{we strayed from } X_0] < -1\frac{1}{2}(1 + \mathbb{E}[\tau_{Z=1}]) = -\frac{1}{2}(1 + \frac{1}{b}) = -\frac{b+1}{2b}$.

In expectation, the number of timesteps that passes between consecutive green regions is $\mathbb{E}[s'] \leq q_u + p_d + q_s(1 + \mathbb{E}[\tau_{Z=1}]) = q_u + p_d + q_s\frac{b+1}{b}$. This means at time $s$ we have, in expectation, visited $\frac{s}{q_u + p_d + q_s\frac{b+1}{b}}$ green nodes.

The expected depth of each consecutive green node is $p_d - 2q_u - q_s\frac{b+1}{2b}$ levels higher than its predecessor. Due to Assumption 3.3 we know that this is strictly positive.

In expectation, the last green node that we have visited is at a depth of $\frac{s}{q_u + p_d + q_s\frac{b+1}{b}}\left(p_d - 2q_u - q_s\frac{b+1}{2b}\right)$. We also know an upper bound on the expected number of steps between green nodes: For any given state $X_s$ of the search algorithm, we know that in expectation, we have taken at most $\mathbb{E}[s'] \leq q_u + p_d + q_s\frac{b+1}{b}$ steps since the last green region. We are interested in a bound of the depth of the current region. In the worst case scenario, all of these steps were backtracks. This leads to $\mathbb{E}[D(X_s)] \geq \frac{s}{q_u + p_d + q_s\frac{b+1}{b}}\left(p_d - 2q_u - q_s\frac{b+1}{2b}\right) - 2(q_u + p_d + q_s\frac{b+1}{b})$ (which is a linear function of $s$). $\square$

*Proof for Lemma 3.9.* Let $x_q = (1 + d)\mathbf{e}$. We denote the probability of the query point inside of $X$ being preferred as $\mathbb{P}[\vec{0} \succ x_q | \mathbf{x}_t] = \mathbb{P}[X \succ F | \mathbf{x}_t]$

We will now show that there are two probabilities $p_X > p_F > 0$ such that:

- $\mathbf{x}_t \in X \implies \mathbb{P}[X \succ F | \mathbf{x}_t] \geq p_X$
- $\mathbf{x}_t \in F \implies \mathbb{P}[X \succ F | \mathbf{x}_t] \leq p_F$

This immediately allows the use of a binomial test. Any level of accuracy is possible, we simply need to repeat the query often enough.

The target location inside $X$ for which $\mathbb{P}[X \succ F | \mathbf{x}_t]$ is smallest is $\mathbf{x}_c = \arg\min_{\mathbf{x}_t \in X} \mathbb{P}[X \succ F | \mathbf{x}_t] = \mathbb{1}$. We call this point $\mathbf{x}_c$ since it lies in a corner of the hypercube. A formal proof that the minimum is found at $\mathbf{x}_c$ is derived with sympy Meurer et al. [2017] and included in the supplementary code. For any parametrization of $\gamma - CKL$ (or another scale-free oracle model) we can now explicitly calculate the lower bound: $p_X = \mathbb{P}[X \succ F | \mathbf{x}_t = \mathbb{1}]$.

We define the following distances:

$$d_c = ||\vec{0} - \mathbf{x}_c|| = \sqrt{d}$$
$$d_{qc} = ||\mathbf{x}_q - \mathbf{x}_c|| = \sqrt{d^2 + d - 1}$$
$$d_q = ||\vec{0} - \mathbf{x}_q|| = d + 1$$

The ratio of distances between $\mathbf{x}_c$ and the two query points is $\frac{||\vec{0} - \mathbf{x}_c||}{||\mathbf{x}_q - \mathbf{x}_c||} = \frac{d_c}{d_{qc}}$. We know that any point $\mathbf{x}'$ that induces the same outcome probability must have the same ratio of distances: $\mathbf{P}[\vec{0} \succ \mathbf{x}_q | \mathbf{x}_t = \mathbf{x}'] = \mathbf{P}[\vec{0} \succ \mathbf{x}_q | \mathbf{x}_t = \mathbf{x}_c] \iff \frac{||\vec{0} - \mathbf{x}'||}{||\mathbf{x}_q - \mathbf{x}'||} = \frac{d_c}{d_{qc}}$.

Out of these points, the one with the least distance to $\vec{0}$ lies on the line segment between $\vec{0}$ and $\mathbf{x}_q$. The point with the largest distance to $\vec{0}$ lies on the ray from $\mathbf{x}_q$ to $\vec{0}$, at $\vec{0} - \mathbf{e}\frac{d + \sqrt{d^3 + d^2 - d}}{d - 1}$. This can be found by solving the condition of equal ratio for the x coordinate. A full derivation in sympy can be found in the supplementary material. We know that all points $\mathbf{x}'$ with a ratio that is strictly larger than $\frac{d_c}{d_{qc}}$ must induce a smaller probability $\mathbf{P}[\vec{0} \succ \mathbf{x}_q | \mathbf{x}_t = \mathbf{x}'] < p_X$. The point farthest away from $\vec{0}$ which still has this ratio lies at $\vec{0} - \mathbf{e}\hat{r}$, with $\hat{r} = \frac{d + \sqrt{d^3 + d^2 - d}}{d - 1}$. This allows us to specify an uncertainty region. Let $r_u = \hat{r} + 1$. The probability $p_F$ can now be explicitly computed (for any parametrization of $\gamma$-CKL or other scale-free oracle models) as $p_F = \mathbf{P}[\vec{0} \succ \mathbf{x}_q | T = \vec{0} - \mathbf{e}(\hat{r} + 1)] < p_X$

$\square$

*Proof for Lemma 3.10.* Let $\mathcal{B}'$ be a hypercube, centered at $\mathbf{x}_t$ and with edge length $2\frac{1}{16} = \frac{1}{8}$. If a cell $c_k$ is in class (A) or (B) then its center $\mathbf{x}_{c_k}$ must lie in the hypercube $B$. We now extend the edge length of this hypercube to fully contain any cell whose center lies in the hypercube. Let $\mathcal{B}$ be a hypercube, centered at $\mathbf{x}_t$ and with edge length $2\frac{1}{16} + r_c$. It follows immediately that $\bigcup_{c_k \text{ has class (B) or (A)}} c_k \subseteq \mathcal{B}$. We have chosen $r_c < \frac{1}{8r_u} < \frac{1}{8}$. Therefore it follows that the edge length of $\mathcal{B}$ is $2\frac{1}{16} + r_c < \frac{1}{8} + \frac{1}{8} = \frac{1}{4}$. $\square$

*Proof for Theorem 3.11.* The tiling $\mathcal{T}(S, r_c)$ contains $K$ cells, this is also the number of hypothesis tests that we conduct. Conditional on $\mathbf{x}_t$, the oracle replies are independent, and therefore the test outcomes are independent. We assume that the probability of error for any one of the tests is $\delta$. The probability of no error occuring across all tests is therefore $(1 - \delta)^K$. We need $\hat{\delta} = 1 - (1 - \delta)^K$. Lemma 3.9 ensures that we can adjust the hypothesis test for any desired probability of error. It is therefore always possible to choose a number of query observations (depending on the dimensionality and the parameters of the choice model) that leads to the desired $\hat{\delta}$.

In the following we assume that all hypothesis tests have provided correct information. This means, that (H) has not been rejected for the cells in class (A) and it has been rejected for the cells in class (C). We create a bounding box $\mathcal{B}$ around all cells for which hypothesis (H) has not been rejected. From Lemma 3.10 we know that this bounding box has an edge length of at most $\frac{1}{4}$. We now look at all possible locations for $\mathbf{x}_t$ and verify that the decision criterion must lead to a correct decision.

**Case 1**, $\mathbf{x}_t \notin (S \cup X)$:

The bounding box can't overlap with $X$. Therefore we backtrack. This is the correct decision.

**Case 2, $\mathbf{x}_t \in X$:**

There is a cell in class (A), which overlaps with $X$. For this cell, the hypothesis (H) has not been rejected. This means that the bounding box must overlap with $X$. Also, we know that the bounding box has an edge length of less than $\frac{1}{4}$. This means that it can overlap with at most 2 of the tiles in $\mathcal{T}(S, 1/4)$. This means that there is a child region in $D(X)$ which fully contains the bounding box. Our decision criterion proceeds to this child. And we know that the bounding box must contain the target (since we're assuming that all hypothesis tests have returned correctly). This ensures that we are proceeding to a green region.

**Case 3, $\mathbf{x}_t \in S$:**

The target is not in the current region, i.e. $X$ is a red region. If the bounding box happens to not overlap with $X$, we backtrack, which is considered a correct decision. If the bounding box happens to overlap with $X$, then we know that there must be a child which fully contains the bounding box. Our decision criterion proceeds to this child region. We also know that the bounding box contains the target. So we are proceeding to a green region. This is a recovery transition, and it is also considered a correct decision.

We have shown that, under the assumption that all hypothesis tests have provided correct information, the decision criterion leads to a correct transition. Our assumption on the hypothesis tests holds with probability $1 - \hat{\delta}$.

If some hypothesis tests are erroneous, then we can see inconsistent behaviour. For example, it is possible that the bounding box is too large, and overlaps with multiple child regions, or overlaps with both $X$ and $\Omega \setminus S$. We assume that in this case, we backtrack. This can be the wrong decision, but it will happen with at most probability $\hat{\delta}$.

$\square$