# OpenReview forum: "Fast Interactive Search under a Scale-Free Comparison Oracle"
_auai.org/UAI/2024/Conference — UAI 2024 poster_

### Official Review · Reviewer_LGiP · 2024-03-05

**Q2-1 Originality-Novelty:** 3
**Q2-2 Correctness-Technical Quality:** 3
**Q2-5 Clarity Of Writing:** 4

**Q1 Summary And Contributions:**

The authors study fast interactive search in the context of comparison queries (whereby one may efficiently check which of two given elements $i$ and $j$ is closer to some target element $t$).

**Q2-3 Extent To Which Claims Are Supported By Evidence:**

4: Excellent: all claims are supported by very convincing evidence (in the form of comprehensive experimental evaluation, rigorous mathematical proofs, detailed (pseudo-)code, precise references, well-motivated and realistic assumptions) and the authors deliver what they promise.

**Q2-4 Reproducibility:**

4: Excellent: key resources (e.g. proofs, code, data) are available and key details (e.g. proof sketches, experimental setup) are comprehensively described for competent researchers to confidently and easily reproduce the main results.

**Q3 Main Strengths:**

The choice of the problem and the nice delivery.

**Q4 Main Weakness:**

None.

**Q5 Detailed Comments To The Authors:**

Overall:
Please expand the captions of ALL figures to be more precise.

Please add "Input" and "Output" to ALL algorithms.
What do they do? :)

Page 1:
Introduction is friendly :)

Can you give some intuition about the low dimensionality of the embeddings?

Page 2:
I find very interesting the psychological aspects that you are referring to.
Could you please elaborate on that?

Page 3:
After Equation (2):
Why does that probability become an indicator function?

Page 4:
Can you please give a high level overview of the convergence analysis first (before giving the details)? :)

Page 5:
Please add some further intuition about your assumptions :)

Is Lemma 3.6 simply a concentration bound?

Page 6:
Lemma 3.9:
Can you please give some intuition about the choice of $r_u$?

Item (B):
Can you please elaborate here? :)

Page 7:
The implementation and empirical evaluation look good :)
(I am not an expert in implementations, but yours look OK.)

Page 8:
Synthetic data:
Can you please give some more details on the generation of synthetic data? :)

Conclusions are very well written.
Can you please elaborate further on future work?

**Q9 Complying With Reviewing Instructions:**

Yes

---

> ### Author Rebuttal · Authors · 2024-04-05
>
> Dear Reviewer,
> Thank you so much for this friendly and very insightful feedback.
> We really appreciate the attention and care that you have invested in reviewing our work.
>
> In particular, your comments on page 4 (highlevel overview of the convergence analysis) and page 6 (choice of $r_u$) are very helpful. If our work is accepted at UAI, we will use the additional space allowed in the camera-ready version to add some additional explanations.
>
> Re detailed comments:
> Page 1-2:
> The $\gamma$-CKL model decouples the error probability from the number of dimensions, and we argue that this additional flexibility is important to fit the oracle (user) as closely as possible to data. Suppose item features for three items $i,j$, and $k$ are given as $x_i$, $x_j$, and $x_k$ (e.g., colors defined via their RGB values). The oracle should have less trouble deciding between i and j the more features it has available (e.g., it is easier to decide which of two colors is closer to a reference color if they are rendered as full RGB, versus just greyscale, say). More generally, the dimension (number of available or learned features per item) and the cognitive difficulty of telling items apart (relative to a reference) should be two independently controllable aspects of the oracle in order to match the behavior of real users/data optimally.
>
> Page 3: Let’s say we look at a triplet $(i,j,k)$ with embedding vectors $x_i$, $x_j$ and $x_k$. The outcome probability is based on a ratio of distances, to the power of $\gamma$: $d_{ij} = ||x_i - x_j||, d_{ik} = ||x_i - x_k||, p = \frac{d_{ij}^\gamma}{d_{ij}^\gamma + d_{ik}^\gamma}$. If $\gamma$ tends to infinity, $p$ tends to 0 or 1.
>
> Page 4 and 5: We will include a high-level sketch of the proof (random walk on the region graph, stochastic coupling, recurrence analysis and bounds) at the beginning of Section 3 and add more details on a real-world implementation at the end. The assumptions, in particular Assumption 3.3, are designed to facilitate the recurrence analysis. But we can give some additional intuition. Importantly: The assumptions are easy to satisfy and do not preclude any practical applications of interest.
>
> Yes, you are correct, Lemma 3.6 is basically a concentration bound. We can add a comment on this to provide more intuition.
>
> Page 6: Our search scheme is scale-invariant. To facilitate notation, we drop scaling factors and discuss our hypothesis test and discretization scheme over a unit hypercube.
> First we present a hypothesis test for the question “Is the target inside region $X$?” The test must reply “yes” or “no”, but if the target lies on the boundary of $X$, the reply is unreliable.
> Therefore we introduce an uncertainty region, which is a sphere of radius $r_u$.
> If the target is inside $X$ we want the test to reliably say “yes”.
> If the target is far outside of $X$, we want the test to reliably say “no”.
> We define “far outside of $X$” in terms of the uncertainty region. Only if the target lies outside of the sphere with radius $r_u$, then the test must recognize with high probability that the target is not in $X$. If the target lies in the uncertainty region around $X$, then we make no guarantees.
>
> The test “Is the target inside of X” becomes more difficult as the dimension $d$ of the embedding space increases. As $d$ grows, the radius of the uncertainty region grows as well. This dependency is presented in Lemma 3.9. We will provide additional comments for this dependence.
>
> Imagine that we cover the current belief region with a discretization T of small cells. For each cell we apply the test. If there is a cell containing the target, then for this cell the test must tell us with high probability “the target is in this cell”. But surrounding this cell will be a cluster of other cells that may incorrectly tell us “the target is in this cell”. We can’t decide which one of these cells is the correct one.
> Item (B) on page 6 describes all cells for which the test may incorrectly tell us “target is in this cell”. For these cells, the target lies inside of the uncertainty region.
> We then discuss a bounding box that must contain the union of all cells that may contain the target. The scaling of the discretization T is chosen in such a way that this bounding box must be fully contained in a child of X. Then we can still pick a child region of X which must, with high probability, contain the target.
>
> Page 8:
> For the synthetic data, we are sampling a target uniformly at random. We then run our search algorithm and simulate random oracle replies (for a range of parametrizations of the $\gamma$-CKL oracle).

---

### Official Review · Reviewer_QXFU · 2024-03-18

**Q2-1 Originality-Novelty:** 3
**Q2-2 Correctness-Technical Quality:** 3
**Q2-5 Clarity Of Writing:** 3

**Q1 Summary And Contributions:**

The authors study the problem of interactive search (active learning) using comparison data of the form "is item x closer to the target than item y?". The authors propose the $\gamma$-CKL model which generalizes the CKL model of Tamuz et al. This class of comparison model is also known as scale-invariant comparison model and exhibits advantageous practical and theoretical properties over some of the other comparison model studied in the literature. Experimental results show that the proposed algorithm and model are competitive to previous approaches in the literature. An additional experiment with real online interactive data showcases the usefulness of the proposed active learning algorithm.

**Q2-3 Extent To Which Claims Are Supported By Evidence:**

3: Good: the main claims are supported by convincing evidence (in the form of adequate experimental evaluation, proofs, (pseudo-)code, references, assumptions).

**Q2-4 Reproducibility:**

3: Good: key resources (e.g. proofs, code, data) are available and key details (e.g. proofs, experimental setup) are sufficiently well-described for competent researchers to confidently reproduce the main results.

**Q3 Main Strengths:**

1. The authors propose a simple generalization of the CKL model of Tamuz et al that addresses an important limitations of the original model related to the 'curse of dimensionality'. This modification is theoretically justified in Theorem 2.1 and is empirically supported by the experiment results shown in Section 4.1.

2. The active learning problem is studied in depth with rigorous theoretical guarantee. The authors show that an intuitive search algorithm can, within a limited number of rounds of comparisons, correctly approximate the target. These are non-trivial results. Combined with the practical usability of the scale-free model, this means that the proposed active learning algorithm can be quite useful in practice. The practical implementation of the active learning algorithm when there is a finite number of samples is described in detail (though in the supplementary materials).

3. The experiment setup is reasonable and showcases the competitiveness of $\gamma$-CKL to the other models in the literature. This corroborates the practical usefulness of the proposed model. The online interactive user-study is quite interesting and clearly demonstrates the superiority of $\gamma$-CKLSearch as an algorithm over the commonly used GaussSearch algorithm for the probit model.

**Q4 Main Weakness:**

1. My impression is that not much of the theoretical guarantee of the continuous setting translates to the real-life data setting with a finite number of data points.

2. It's not clear to me what the intuition behind Assumption 3.3 is. It seems rather arbitrary and seemingly used to ensure that the resulting analysis of Algorithm 1 goes through.

3. Section 3.3 should go into more details describing the practical implementation of the $\gamma$-CKL search algorithm in the real life setting where we only observe a finite number of data points instead of delegating the descriptions to the supplementary materials.

**Q5 Detailed Comments To The Authors:**

See points about weaknesses. Overall, I think these are still minor weaknesses that can be readily addressed.

**Q9 Complying With Reviewing Instructions:**

Yes

---

> ### Author Rebuttal · Authors · 2024-04-05
>
> Dear Reviewer,
> We thank you for your time in evaluating our submission, and we are grateful for your discerning and helpful comments.
> Please find below our responses addressing the questions raised in your review. If the paper is accepted, we plan to revise the manuscript accordingly.
>
> Q4.1
> In scenarios with a finite number of items, we believe the theory gives us the following insight. Starting with a large number of items, we expect the situation to be similar to the dense case we study theoretically, and informally we expect the algorithm to be able to “zoom in” on the target with an exponential rate of convergence, until it has arrived at a zoom level at which the dataset begins to look sparse. At that point, the theory is no longer applicable, but we expect to have filtered the search space down to a small number of items, such that identifying the target object among the remaining items is much easier to do.
>
>
>
> Q4.2
> This is a valid observation. Assumption 3.3 is indeed an artifact of the analysis of Algorithm 1. We’re presenting a random walk on a graph that is biased towards the right solution. In practice, we find that this bias needs to be strictly positive, Assumption 3.3 is designed to facilitate the recurrence analysis.
> Nevertheless, Assumption 3.3 is simple to satisfy and does not prevent any practical applications. We already include a constructive proof for a scheme which must always satisfy Assumption 3.3.
> We very much appreciate this feedback and will improve the clarity of our writing. We’ll provide additional intuition on the choice of Assumption 3.3 and it’s role in the overarching proof approach.
>
> Q4.3
> Our search scheme has an intuitive and appealing implementation for a real-life setting.
> We would be happy to use the additional space allowed for a camera-ready version to move some of these descriptions into the main body of the paper.

---

### Official Review · Reviewer_jkt8 · 2024-03-22

**Q2-1 Originality-Novelty:** 2
**Q2-2 Correctness-Technical Quality:** 3
**Q2-5 Clarity Of Writing:** 2

**Q1 Summary And Contributions:**

The paper presents a new probabilistic oracle model that can be used for searching a target item in a database by answering a sequence of queries of the form (i, j, t) meaning which of the items i and j is closer to item t. Specifically, the paper proposes a simple extension of an existing triplet model called CKL, analyses the theoretical properties of the proposed method, and validates the method empirically on a collection of benchmarks known in the community. The experimental evaluation includes a user study and the experimental results demonstrate clearly the performance of the proposed method compared with existing approaches.

**Q2-3 Extent To Which Claims Are Supported By Evidence:**

2: Fair: the main claims are somewhat supported by evidence (but the experimental evaluation may be weak, or does not match entirely with the claims, important baselines may be missing, proofs contain important ideas but lack rigor, algorithmic details are only discussed superficially, references are imprecise, assumptions are not sufficiently motivated or explicated, etc.).

**Q2-4 Reproducibility:**

2: Fair: key resources (e.g. proofs, code, data) are unavailable but key details (e.g. proof sketches, experimental setup) are sufficiently well-described for an expert to confidently reproduce the main results.

**Q3 Main Strengths:**

The paper considers an important problem in information retrieval with potentially many different real-world applications.

The empirical evaluation is sound and also includes a user study. The results are presented in a fairly clear manner and therefore it is easy to understand the performance of the proposed method.

**Q4 Main Weakness:**

I found the novelty of the proposed method fairly limited. In addition, it is hard to follow the convergence analysis from section 3.1 without a more concrete example. I didn't find figures 1 and 2 particularly useful in following the details. Perhaps it's better to start from a simpler, more intuitive example which will definitely improve the quality of the presentation.

As far as I understood, the underlying iterative search procedure assumes a finite number of queries. So, is it possible to bound the number of queries as a function of the \gamma parameter?

**Q5 Detailed Comments To The Authors:**

Theorem 1 establishes a linear dependency between \gamma and the number of dimensions d. I suppose this is fine from a theoretical perspective but in practice how do we choose the value of \gamma. The empirical evaluation doesn't seem to shed light on this issue.

**Q9 Complying With Reviewing Instructions:**

Yes

---

> ### Author Rebuttal · Authors · 2024-04-05
>
> Dear Reviewer,
>
> Thank you for your feedback. We would like to respond to your specific question Q5 after commenting on the broader contribution of the paper in response to Q4.
>
> This paper's contribution is the scale-free oracle model, but also the algorithm for interactive search adapted for this model. To the best of our knowledge, this work is the first to describe a probabilistic interactive search algorithm with provably exponential convergence. To show this, we analyze an induced random walk on a graph over search regions. A recurrence analysis based on a stochastic coupling allows us to formally prove exponential bounds on the rate of convergence. This is different from existing work on probabilistic oracle models. We support our model and algorithm with an extensive empirical analysis.
>
> A bound on the number of queries needed to find an item among a finite set is difficult, because this bound would depend on the locations of all the items in the feature space. For this reason, for the formal analysis, we assume that items are dense within the feature space, so that the target and query items may be at any location. This allows us to reason about the speed of convergence of the search process towards the target, instead of a stopping time of that process over a finite set of items. This is a non-trivial result, since the search is based on a probabilistic oracle and must be able to recover from inevitable “wrong turns”.
> In a scenario where only finitely many items are available, the algorithm will “zoom in” on the target with an exponential rate of convergence, until it terminates at a zoom level at which items are sparse enough so as to leave only the destination in the belief region.
> We believe that many real-world datasets are large enough (millions of items or more) for the speed of convergence to matter (music, video clips, posts on social media). We will make this point more explicitly in the theoretical analysis in Section 3.
>
> Regarding the detailed comment in Q5:
> Theorem 2.1 provides insight into the dependency between $\gamma$, the dimension $d$, and the behavior of the oracle. When learning an embedding, $\gamma$ (and an unknown dimension $d$) can be considered a hyperparameter of the choice model and should be adapted to each new dataset. We present empirical results based on cross-validation in Section 4.

---

### Official Review · Reviewer_o1K2 · 2024-03-24

**Q2-1 Originality-Novelty:** 2
**Q2-2 Correctness-Technical Quality:** 3
**Q2-5 Clarity Of Writing:** 3

**Q10 Ethical Concerns:**

No.

**Q1 Summary And Contributions:**

The paper presents an oracle model for interactive search within a database. In each query, the user specifies which of two given items better matches the target item. This model distinguishes itself from others in the literature due to its scale-free nature, enabling an exponentially rapid reduction of the belief region. It also includes empirical results to support the proposed oracle model.

**Q2-3 Extent To Which Claims Are Supported By Evidence:**

3: Good: the main claims are supported by convincing evidence (in the form of adequate experimental evaluation, proofs, (pseudo-)code, references, assumptions).

**Q2-4 Reproducibility:**

3: Good: key resources (e.g. proofs, code, data) are available and key details (e.g. proofs, experimental setup) are sufficiently well-described for competent researchers to confidently reproduce the main results.

**Q3 Main Strengths:**

1. The rapid halving of the belief region represents a significant advancement compared to previous studies.

**Q4 Main Weakness:**

1. Does scale-free mean the computational costs of Algorithm 1 is independent of d (i.e., the embedding dimension)?

**Q5 Detailed Comments To The Authors:**

See Q4.

**Q9 Complying With Reviewing Instructions:**

Yes

---

> ### Author Rebuttal · Authors · 2024-04-05
>
> We thank you for your time in evaluating our submission and constructive comments.
>
> > Does scale-free mean the computational costs of Algorithm 1 is independent of d (i.e., the embedding dimension)?
>
> The concept of a scale-free or scale-invariant oracle model refers to the property that the probability assigned to a triplet comparison $(i, j, k)$ does not depend on the _absolute_ distances between the corresponding embedding vectors $(x_i, x_j, x_k)$, but only on the _relative_ distances. If all three vectors were to be scaled by a factor a, leading to $(a x_i, a x_j, a x_k)$, the outcome probability would not change. To your point: the scale-free property relates to the query complexity, and does not speak to the computational costs and their dependence on $d$.
>
> An important consequence of the scale-free property is that, informally, the number of queries that are necessary to halve the search space is constant and does not depend on the initial volume of the space.

---

### Official Review · Reviewer_GY4b · 2024-03-31

**Q2-1 Originality-Novelty:** 3
**Q2-2 Correctness-Technical Quality:** 3
**Q2-5 Clarity Of Writing:** 2

**Q1 Summary And Contributions:**

This paper proposes a scale-free probabilistic oracle model, \gamma-CKL, that can be used for target item search problem settings. To answer whether item i is closer than item j to target item t, the idea is to essentially use ratio of distances powered by a reasonable parameter \gamma. The key result that the authors show is that for larger dimensions, \gamma-CKL is a better choice with higher values of \gamma than existing methods. The authors also show an exponential rate of convergence for their proposed search algorithm. Experiments on simulated and real-world datasets show the efficacy of the proposed method. The interesting part is the user study, where the authors have recruited real users to search the target item using such queries.

**Q2-3 Extent To Which Claims Are Supported By Evidence:**

3: Good: the main claims are supported by convincing evidence (in the form of adequate experimental evaluation, proofs, (pseudo-)code, references, assumptions).

**Q2-4 Reproducibility:**

3: Good: key resources (e.g. proofs, code, data) are available and key details (e.g. proofs, experimental setup) are sufficiently well-described for competent researchers to confidently reproduce the main results.

**Q3 Main Strengths:**

1. Relevant problem for the community
2. A reasonable extension that leads to exponential rate of convergence is an interesting contribution
3. Real user study results are plus to the paper

**Q4 Main Weakness:**

The presentation of the theoretical parts can be improved. It required me to read that part of the paper several times.

**Q5 Detailed Comments To The Authors:**

The following are some questions on which the authors can rebut on:

1. What is a CKL method in abstract? The authors should define the abbreviation in the abstract. In fact, the authors use this abbreviation a lot in the paper but is never explained.

2. In related work section, it is not clear how Tamuz et al.’s work is different from the current work. What does \gamma=2 signify? Till this point in the paper, it is not clear what is even \gamma?

3. Assumption 3.2 onwards, it is difficult to follow the paper. I would recommend just keeping the main theorems in the paper and move all supplementary lemmas in the appendix. Instead of Lemmas, rather discuss the intuition of the proof.

4. Page 7, right column, first paragraph: I am not sure how from Figure 4, one can conclude that higher the values of d, better the performance by higher \gamma values. Can the authors clarify? Also, how is \gamma chosen for these experiments? Is it cross-validated along with other parameters.

5. Practically, the queries in section 4.2 are different from the triplet-based queries studied in the paper. Are there implications because of these modifications? Also, what are the values of d and \gamma used here? Were there any experiments to verify theorem 2.1 for the interactive user study?

**Q9 Complying With Reviewing Instructions:**

Yes

---

> ### Author Rebuttal · Authors · 2024-04-05
>
> Dear Reviewer,
> We thank you for your time in evaluating our submission, and we are grateful for your discerning and helpful comments.
> Please find below our responses addressing the questions raised in your review. If the paper is accepted, we plan to revise the manuscript accordingly.
>
> 1) CKL is an abbreviation for Crowd Kernel Learning, it was published in “Adaptively Learning the Crowd Kernel” by Tamuz et al, cited in the Introduction section.
>
> 2) The Crowd Kernel Learning method is an embedding algorithm that learns a feature vector for each item in a dataset of triplet comparison outcomes. The authors use a probabilistic approach and optimize the log-likelihood of comparison outcomes given a triplet oracle model. To the best of our knowledge, Tamuz et al are the first to propose a scale-free oracle model: The likelihood of a triplet comparison outcome is based on the ratio of _squared_ l2 distances between the corresponding embedding vectors.
> We present an extension to this choice model by introducing a $\gamma$ parameter. The choice probabilities of $\gamma$-CKL are based on a ratio of l2 distances to the power of $\gamma$. For $\gamma=2$ we recover the original CKL model.
>
>     We will update the related works section to clarify that our research is fundamentally different from “Adaptively Learning the Crowd Kernel”. Tamuz et al focus on an embedding algorithm and only briefly discuss the potential applications in search. We propose a novel framework for an interactive search scheme with exponential convergence guarantees based on the introduced $\gamma$-CKL model.
>
> 3) This is a difficult trade-off. We have already delegated much of the formalism to the appendix by only including proof sketches in the main paper. We do want to highlight the tools and concepts that enable the convergence analysis, i.e., a biased random walk on a graph, stochastic coupling, and recurrence analysis. Nevertheless, we appreciate the feedback and will consider if we can further condense Section 3.1. For example, we could move Lemma 3.6 to the appendix.
>
> 4) We empirically find that for higher dimensions $d$, higher values of $\gamma$ perform better for all 3 datasets (which aligns well with the theory we present in Section 2 and the appendix). We will include additional plots, best performing $\gamma$ versus growing $d$, for each dataset to support our claim in the main text.
> Yes, $\gamma$ is cross-validated along with other parameters.
>
>
> 5) The purpose of this user study is to provide a reference point comparing the efficiency of our new choice model relative to an established search algorithm. It is challenging to produce such a comparison. The interactive search setting requires a human participant in the loop. Because the algorithm generates queries sequentially, it is not possible to conduct such a study based on an offline dataset. The most relevant experimental study is described by Chumbalov et al. We have therefore replicated the exact setup of their evaluation.
> We believe that the favorable empirical query-complexity we find in comparison to GaussSearch (the competing search algorithm by Chumbalov et al.) provides some evidence in favor of the $\gamma$-CKL model. Our findings reproduce the performance of Gausssearch as recorded by Chumbalov et al. and show a statistically significant performance increase when using our novel scale-free approach. The hyperparameters we use in Section 4.2 are $\gamma = 3$, $D = 5$ (see Appendix B for details).

---

### Meta-Review · Area_Chair_UxYs · 2024-04-21

Most reviewers recommended acceptance whereas one reviewer recommended rejection. The reviewers did not respond to the authors' rebuttal nor did they engage in the meta-reviewer/reviewer discussion.

I have gone through the reviews and the authors' rebuttal. I did not find the critical reviewer's points convincing for rejection. The agreement between most reviewers seems to be that the paper provides a non-trivial extension of the prior CKL method. The theory seems sound and experiments sufficient.